# Structures of ferroportin in complex with its specific inhibitor vamifeport

Elena Farah Lehmann[1†], Márton Liziczai[1†], Katarzyna Drożdżyk[1],
Patrick Altermatt[2], Cassiano Langini[1], Vania Manolova[2], Hanna Sundstrom[2],
Franz Dürrenberger[2], Raimund Dutzler[1*], Cristina Manatschal[1*]

[1]Department of Biochemistry, University of Zurich, Zürich, Switzerland; [2]CSL Vifor, St. Gallen, Switzerland

**Abstract** A central regulatory mechanism of iron homeostasis in humans involves ferroportin (FPN), the sole cellular iron exporter, and the peptide hormone hepcidin, which inhibits $Fe^{2+}$ transport and induces internalization and degradation of FPN. Dysregulation of the FPN/hepcidin axis leads to diverse pathological conditions, and consequently, pharmacological compounds that inhibit FPN-mediated iron transport are of high clinical interest. Here, we describe the cryo-electron microscopy structures of human FPN in complex with synthetic nanobodies and vamifeport (VIT-2763), the first clinical-stage oral FPN inhibitor. Vamifeport competes with hepcidin for FPN binding and is currently in clinical development for β-thalassemia and sickle cell disease. The structures display two distinct conformations of FPN, representing outward-facing and occluded states of the transporter. The vamifeport site is located in the center of the protein, where the overlap with hepcidin interactions underlies the competitive relationship between the two molecules. The introduction of point mutations in the binding pocket of vamifeport reduces its affinity to FPN, emphasizing the relevance of the structural data. Together, our study reveals conformational rearrangements of FPN that are of potential relevance for transport, and it provides initial insight into the pharmacological targeting of this unique iron efflux transporter.

**\*For correspondence:**
dutzler@bioc.uzh.ch (RD);
c.manatschal@bioc.uzh.ch (CSL)

†These authors contributed equally to this work

## Editor's evaluation

This important study reports cryo-EM structures of human ferroportin (FPN), a protein essential for iron transport in humans, and will be of interest to researchers studying membrane transport mechanisms as well as to those interested in drug design. Structures detail interactions between FPN and the small-molecule inhibitor vamifeport, which is currently in clinical trials for sickle cell disease. In addition, they identify a new (occluded) protein conformation, stabilized by a small-protein binder, that may be relevant to the transport mechanism. Evidence for the mechanism of inhibition by vamifeport is convincing.

## Introduction

The ability of iron to alter its oxidation state and to form coordinative interactions with free electron pairs makes it an essential cofactor of numerous proteins involved in the catalysis of redox reactions and the transport of oxygen. However, its reactivity is harmful if present in excess and distributed inappropriately (*Galaris et al., 2019*). Since a regulated excretion pathway of iron does not exist in mammals, its uptake, recycling, and redistribution are tightly regulated processes (*Nemeth and Ganz, 2021*). Dysregulated iron homeostasis leads to anemia or iron overload and has also been linked to numerous other disorders such as cancers, neurodegenerative and age-related diseases (*Crielaard et al., 2017*). Consequently, the targeting of proteins involved in iron metabolism constitutes a

promising but somewhat underexplored pharmacological strategy for the treatment of such disorders (**Crielaard et al., 2017**).

While iron can be imported into cells as either $Fe^{2+}$ or $Fe^{3+}$ via several mechanisms involving transporters of the SLC11/NRAMP family, heme transporters and by transferrin-mediated endocytosis, cellular iron export solely proceeds via Ferroportin (FPN), a divalent metal ion transporter present in most cells, with high expression levels found in enterocytes of the duodenum, hepatocytes, and macrophages (**Ganz, 2013**; **Montalbetti et al., 2013**). Consequently, a central mechanism to control the systemic iron concentration proceeds via the regulation of FPN levels at the cell surface. This is accomplished by the peptide hormone hepcidin, which comprises 25 amino acids (**Nemeth et al., 2004**). Hepcidin binds FPN on its extracellular side, thereby reducing iron export by inhibition of the transporter and by promoting its ubiquitination, leading to internalization and degradation (**Aschemeyer et al., 2018**; **Billesbølle et al., 2020**; **Pan et al., 2020**; **Qiao et al., 2012**).

The transport properties of FPN have been studied using iron efflux assays in *Xenopus laevis* oocytes and mammalian cells for the human ortholog (hsFPN) (**Deshpande et al., 2018**; **Manolova et al., 2019**; **Mitchell et al., 2014**) and by proteoliposome-based in-vitro assays for hsFPN, its ortholog from the primate Philippine tarsier (tsFPN) and for a distant bacterial homolog from *Bdellovibrio bacteriovorus* (bbFPN; **Billesbølle et al., 2020**; **Bonaccorsi di Patti et al., 2015**; **Pan et al., 2020**; **Taniguchi et al., 2015**). These studies revealed that extracellular $Ca^{2+}$ stimulates iron transport (**Deshpande et al., 2018**) and that besides $Fe^{2+}$, FPN also transports other transition metal ions including $Co^{2+}$, $Zn^{2+}$, and $Ni^{2+}$ (**Mitchell et al., 2014**). Its transport mechanism as an electroneutral $H^+/Me^{2+}$ antiporter likely facilitates cellular export of the positively charged substrate ion despite the negative resting membrane potential (**Pan et al., 2020**).

Insight into the molecular mechanisms of iron transport and its regulation by hepcidin was provided by various structures of the prokaryotic bbFPN and its mammalian homologs tsFPN and hsFPN (**Billesbølle et al., 2020**; **Deshpande et al., 2018**; **Pan et al., 2020**; **Taniguchi et al., 2015**). FPN belongs to the major facilitator superfamily (MFS), whose members translocate their substrates by traversing between conformations where the substrate binding site, commonly located in the center of the transporter, is alternately accessible from either side of the membrane (**Abramson et al., 2003**; **Drew et al., 2021**). The transition between outward- and inward-open conformations proceeds via occluded states, where the substrate is shielded from the surrounding aqueous environment from both sides. These conformational rearrangements involve the concerted movement of structurally similar N- and C-terminal domains (termed N- and C-domain), each comprising a bundle of six transmembrane α-helices that are both oriented in the same direction with respect to the membrane (**Drew et al., 2021**).

The bacterial homolog bbFPN was crystallized in outward- and inward-facing conformations in the absence and presence of divalent metal ions, thus providing insight into the conformational transitions during transport (**Deshpande et al., 2018**; **Taniguchi et al., 2015**). The cryo-electron microscopy (cryo-EM) structures of hsFPN and tsFPN have revealed structures of the apo-state of the transporter and of its complexes with $Co^{2+}$ and hepcidin in outward-facing conformations (**Billesbølle et al., 2020**; **Pan et al., 2020**). Together, these studies suggest that, distinct from other MFS-transporters, FPN likely contains two conserved substrate binding sites. The first site resides within the N-domain (S1) and comprises an aspartate and a histidine on α1 (i.e. Asp39 and His 43) in both hsFPN and tsFPN. A metal ion binds nearby, between α1 and α6, in the prokaryotic bbFPN. A second site that is confined to mammalian transporters is contained within the C-domain (S2) and bridges a conserved cysteine (Cys 326) on α7 with a histidine on α11 (i.e. His 507 in hsFPN and His 508 in tsFPN). Compared to other MFS transporters, α7 comprises an unusually long unwound part of around 10 residues in its center. Notably, hepcidin protrudes deeply into the outward-facing pocket, thereby occluding the iron efflux pathway and preventing conformational transitions, which explains its inhibitory activity (**Billesbølle et al., 2020**; **Pan et al., 2020**). Moreover, the carboxy-terminus of hepcidin was found to coordinate the bound metal ion at the S2 site, which might enable hepcidin to selectively target FPN engaged in iron transport (**Billesbølle et al., 2020**).

Owing to the central role of hepcidin in regulating iron homeostasis, compounds that either mimic or block its function are of high therapeutic interest, and several agents are currently in clinical development for the treatment of anemia or iron overload conditions in different disorders (**Casu et al., 2018**; **Crielaard et al., 2017**; **Katsarou and Pantopoulos, 2018**). Among those agents, the first orally available FPN inhibitor vamifeport (VIT-2763) is now in clinical trials for the treatment of β-thalassemia

and sickle cell disease (*Manolova et al., 2019*; *Nyffenegger et al., 2022*). On the molecular level, vamifeport was shown to compete with hepcidin for FPN binding with an $IC_{50}$ value in the low nanomolar range (*Manolova et al., 2019*). Moreover, similar to hepcidin, vamifeport induced FPN ubiquitination, internalization, and degradation, albeit to a lower extent and with slower kinetics compared to the peptide hormone. In cellular efflux assays, vamifeport blocked iron export by FPN in a dose-dependent manner with a potency comparable to hepcidin (*Manolova et al., 2019*).

To shed light on the structural details of inhibitor interactions, we have determined cryo-EM structures of hsFPN in complex with vamifeport and synthetic nanobodies. The structures reveal outward-facing and occluded states of hsFPN with vamifeport located in the center of the transporter in vicinity of the S2 substrate site. The binding site is flanked by two hydrophobic pockets that accommodate the terminal aromatic moieties of the inhibitor. One of these pockets overlaps with the known site of hepcidin, which explains the competitive relationship between the two molecules. Mutations in the observed binding site reduce the affinity of vamifeport to hsFPN, validating the results from our structural data. Together, our results provide insight into the interaction of hsFPN with a compound of promising therapeutic potential.

## Results

### Characterization of the hsFPN – vamifeport interaction

To characterize the interaction between vamifeport and hsFPN, we have transiently expressed the transporter in HEK293 cells and found it to be monodisperse and stable when purified in the detergent N-dodecyl-β-D-Maltoside (DDM) with an apparent melting temperature ($T_m$) of 47°C as determined

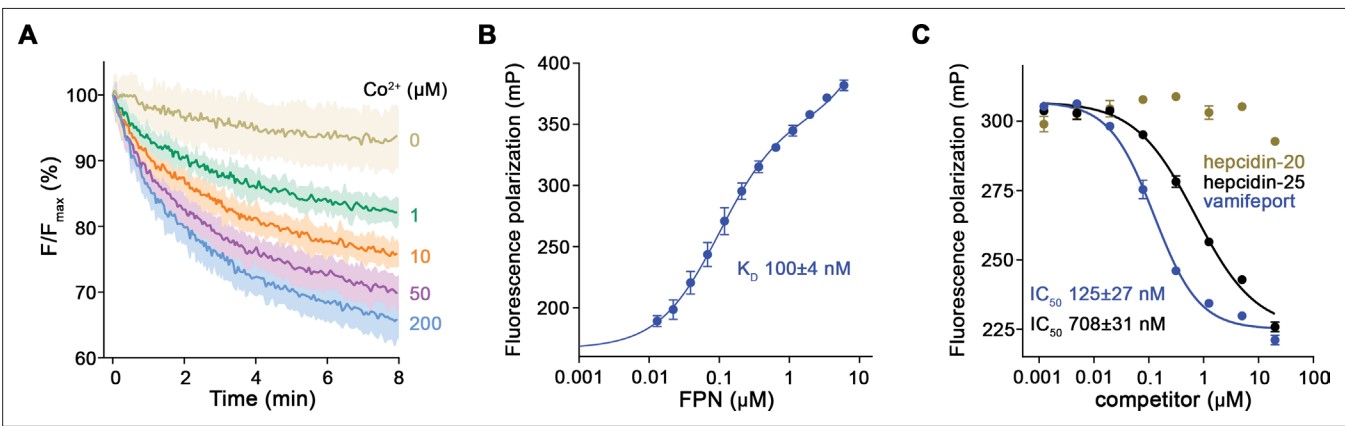

**Figure 1.** Functional characterization of hsFPN. (**A**) Fluorescence-based uptake of $Co^{2+}$ into proteoliposomes containing reconstituted hsFPN. Metal ion influx is monitored with the fluorophore calcein that is trapped inside the liposomes (six measurements for 0 µM $Co^{2+}$, 1 µM $Co^{2+}$, and 10 µM $Co^{2+}$ and eight measurements for 50 µM $Co^{2+}$ and 200 µM $Co^{2+}$ from four independent reconstitutions). (**B**) Binding of fluorescently labeled hepcidin-25 (TMR-hepcidin) to increasing concentrations of hsFPN as monitored by the change in the fluorescence polarization of the peptide (six measurements from two independent experiments). (**C**) Competition of bound TMR-hepcidin with vamifeport and unlabeled hepcidin-20 and hepcidin-25 (six measurements for vamifeport and hepcidin-25 from three independent experiments and four measurements for hepcidin-20 from two independent experiments). (**A–C**) Data show mean of the indicated number of measurements, and errors are SEM.

The online version of this article includes the following source data and figure supplement(s) for figure 1:

**Source data 1.** Proteoliposome-based in-vitro transport data of hsFPN, TMR-hepcidin fluorescence polarization direct binding and displacement data of hsFPN.

**Figure supplement 1.** Biochemical properties of purified hsFPN.

**Figure supplement 1—source data 1.** Size-exclusion chromatography data of hsFPN and corresponding SDS-PAGE gels, thermal stability data, and proteoliposome-based in-vitro transport data of empty liposomes.

**Figure supplement 2.** Characterization of sybodies binding hsFPN.

**Figure supplement 2—source data 1.** Size-exclusion chromatography data of all sybodies and corresponding SDS-PAGE gels, surface plasmon resonance (SPR) binding data of all sybodies.

**Figure supplement 3.** Biochemical properties of sybody-hsFPN complexes.

**Figure supplement 3—source data 1.** Size-exclusion chromatography data of sybody-hsFPN complexes and corresponding SDS-PAGE gels.

in a thermal stability assay (*Figure 1—figure supplement 1A, B*). When reconstituted into liposomes and assayed with the metal ion sensitive fluorophore calcein, hsFPN displayed a robust $Co^{2+}$ transport activity (*Figure 1A*, *Figure 1—figure supplement 1C*) with an apparent $K_M$ in the low µM range. In presence of vamifeport, the apparent melting temperature ($T_m$) of hsFPN is increased by about 5°C, reflecting the stabilizing effect of the inhibitor (*Figure 1—figure supplement 1B*). To further assess the binding properties of vamifeport toward purified hsFPN, we exploited its capability to displace TMR-hepcidin, a fluorescent derivative of the peptide hormone (*Dürrenberger et al., 2013*). The quantification of TMR-hepcidin binding by titrating hsFPN to a constant amount of the labeled peptide and monitoring the change in fluorescence polarization (FP), yielded a $K_D$ of 100±4 nM (*Figure 1B*). The incomplete saturation at high hsFPN concentrations reflects unspecific binding of TMR-hepcidin with one order of magnitude lower affinity. Subsequently, we performed displacement experiments with unlabeled hepcidin and vamifeport. As expected, bound TMR-hepcidin could be readily displaced by hepcidin-25 in a dose-dependent manner with an $IC_{50}$ of ca 700 nM, while hepcidin-20, lacking the first 5 N-terminal amino acids necessary for FPN binding, was inactive (*Figure 1C*). Vamifeport displaced TMR-hepcidin with higher potency with an $IC_{50}$ value of ca 130 nM (*Figure 1C*). To convert obtained $IC_{50}$ values to $K_D$s, we fitted our displacement data to a competition binding model, yielding $K_D$ values of around 150 nM for hepcidin-25 and 25 nM for vamifeport. Taken together, these data show that our purified hsFPN is stable and functional and that it competitively binds hepcidin and vamifeport with $K_D$ values in the nanomolar range.

## Selection and characterization of synthetic nanobodies targeting human FPN

To resolve the molecular details of the interaction between hsFPN and vamifeport, we proceeded with the structural characterization of complexes containing both components. Although single particle cryo-EM has successfully been used to determine high-resolution structures of membrane proteins in complex with comparably small ligands like vamifeport (*de Oliveira et al., 2021*), hsFPN with a molecular weight of only 65 kDa and lacking cytosolic or extracellular domains is of insufficient size for proper particle alignment during the 2D and 3D classification. We therefore set out to generate binding proteins as a strategy to circumvent the described limitations. To this end, we performed a selection of nanobodies from synthetic libraries (sybodies) optimized for membrane proteins (*Zimmermann et al., 2020*). This process has led to the identification of 19 unique binders that recognized hsFPN based on ELISA, five of which showed promising biochemical properties in terms of expression levels and aggregation behavior (*Figure 1—figure supplement 2A, B*). Binding affinities were subsequently quantified using surface plasmon resonance (SPR) measurements. While the sybodies 1, 8, and 11 interacted with hsFPN with low-micromolar affinity, sybodies 3 and 12 (Sb3$^{FPN}$ and Sb12$^{FPN}$, short Sy3 and Sy12) bound hsFPN with $K_D$s in the high-nanomolar range (i.e. 500 nM for Sy3 and 308 nM for Sy12; *Figure 1—figure supplement 2C–G*). Binding of the latter two sybodies to detergent-solubilized hsFPN resulted in a complex that eluted as monodisperse peak during size-exclusion chromatography (SEC) at a similar elution volume as hsFPN alone (*Figure 1—figure supplement 3A, B*). Both sybody complexes also remained intact and co-eluted with hsFPN on SEC after the incubation with an excess of vamifeport, indicating that the interaction with the compound did not interfere with their binding (*Figure 1—figure supplement 3A, B*).

## hsFPN structures in complex with Sy3 and Sy12

For structural studies, we prepared complexes of hsFPN with either Sy12 or Sy3 in the presence of vamifeport and in the latter case also in the absence of the compound and collected cryo-EM data that allowed the reconstruction of cryo-EM maps at 3.9, 3.2, and 4.1 Å resolution, respectively (*Figure 2—figure supplements 1–3*, *Table 1*). In the obtained structures, both sybodies bind to distinct epitopes on the extracellular side (*Figure 2A and B*). The isotropic distribution of particles in the data of hsFPN/Sy3 complexes has yielded high-quality maps for both samples that are well defined for the entire structure but better resolved in the vamifeport complex (*Figure 2B*, *Figure 2—figure supplements 2–4*). In contrast, the quality of the map of the hsFPN/Sy12/vamifeport complex is to some degree compromised by a preferential orientation of the particles and difficulties to reconstruct and refine a 3D model as a consequence of the binding of Sy12 to a flexible loop region at the periphery of the protein (*Figure 2A*, *Figure 2—figure supplement 1*). Still, even in latter case,

**Table 1.** Cryo-electron microscopy (cryo-EM) data collection, refinement, and validation statistics.

| | Dataset 1<br>hsFPN/Sy3<br>(EMDB-16353)<br>(PDB 8C02) | Dataset 2<br>hsFPN/Sy3-vamifeport<br>(EMDB-16345)<br>(PDB 8BZY) | Dataset 3<br>hsFPN/Sy12-vamifeport<br>(EMDB-16354)<br>(PDB 8C03) |
|---|---|---|---|
| **Data collection and processing** | | | |
| Microscope | FEI Titan Krios | FEI Titan Krios | FEI Titan Krios |
| Camera | Gatan K3 GIF | Gatan K3 GIF | Gatan K3 GIF |
| Magnification | 130,000 | 130,000 | 130,000 |
| Voltage (kV) | 300 | 300 | 300 |
| Electron exposure (e–/Å$^2$) | 66 | 61/70.795 | 69.559/70.766 |
| Defocus range (μm) | –2.4 to –1 | –2.4 to –1 | –2.4 to –1 |
| Pixel size (Å) | 0.651 (0.3255) | 0.651 (0.3255) | 0.651 (0.3255) |
| Symmetry imposed | C1 | C1 | C1 |
| Initial particle images (no.) | 1,927,334 | 1,820,142 | 3,045,873 |
| Final particle images (no.) | 139,011 | 320,701 | 387,990 |
| Map resolution (Å)<br>FSC threshold 0.143 | 4.09 Å | 3.24 Å | 3.89 Å |
| Map sharpening b-factor (Å$^2$) | –201.2 | –160.5 | –202.1 |
| **Refinement** | | | |
| Model resolution (Å)<br>FSC threshold 0.5 | 4.3 | 3.4 | 4.1 |
| Non-hydrogen atoms | 4245 | 4314 | 3331 |
| Protein residues | 549 | 549 | 429 |
| Ligand | 0 | 2 | 1 |
| Water | 0 | 0 | 0 |
| B factors (Å$^2$) protein | 162.30 | 168.28 | 130.63 |
| B factors (Å$^2$) ligand | --- | 116.10 | 63.72 |
| RMSD Bond lengths (Å) | 0.004 (0) | 0.005 (0) | 0.004 (0) |
| RMSD Bond angles (°) | 0.634 (0) | 0.732 (0) | 0.645 (0) |
| MolProbity score | 1.88 | 1.63 | 2.00 |
| Clashscore | 10.16 | 7.02 | 6.37 |
| Poor rotamers (%) | 1.10 | 1.10 | 4.46 |
| CaBLAM outliers (%) | 2.59 | 1.31 | 0.96 |
| Ramachandran outliers (%) | 0.00 | 0.00 | 0.00 |
| Ramachandran allowed (%) | 4.64 | 3.33 | 2.84 |
| Ramachandran favored (%) | 95.36 | 96.67 | 97.16 |

the map was sufficiently well resolved to permit its interpretation by an atomic model for the bulk of the transporter (*Figure 2A*, *Figure 2—figure supplements 1 and 4*). As defined previously, hsFPN consists of two topologically related domains with equivalent orientation in the membrane (*Billesbølle et al., 2020*; *Pan et al., 2020*; *Figure 2C*, *Figure 2—figure supplement 5*). The N-domain comprises α1–α6 and the C-domain α7–α12 with α7 being unwound in the center and preceded by an amphipathic helix that is oriented parallel to the intracellular membrane plane (*Figure 2C*). For both complexes, the respective N- and C-termini (about 20–30 residues each), the extended intracellular

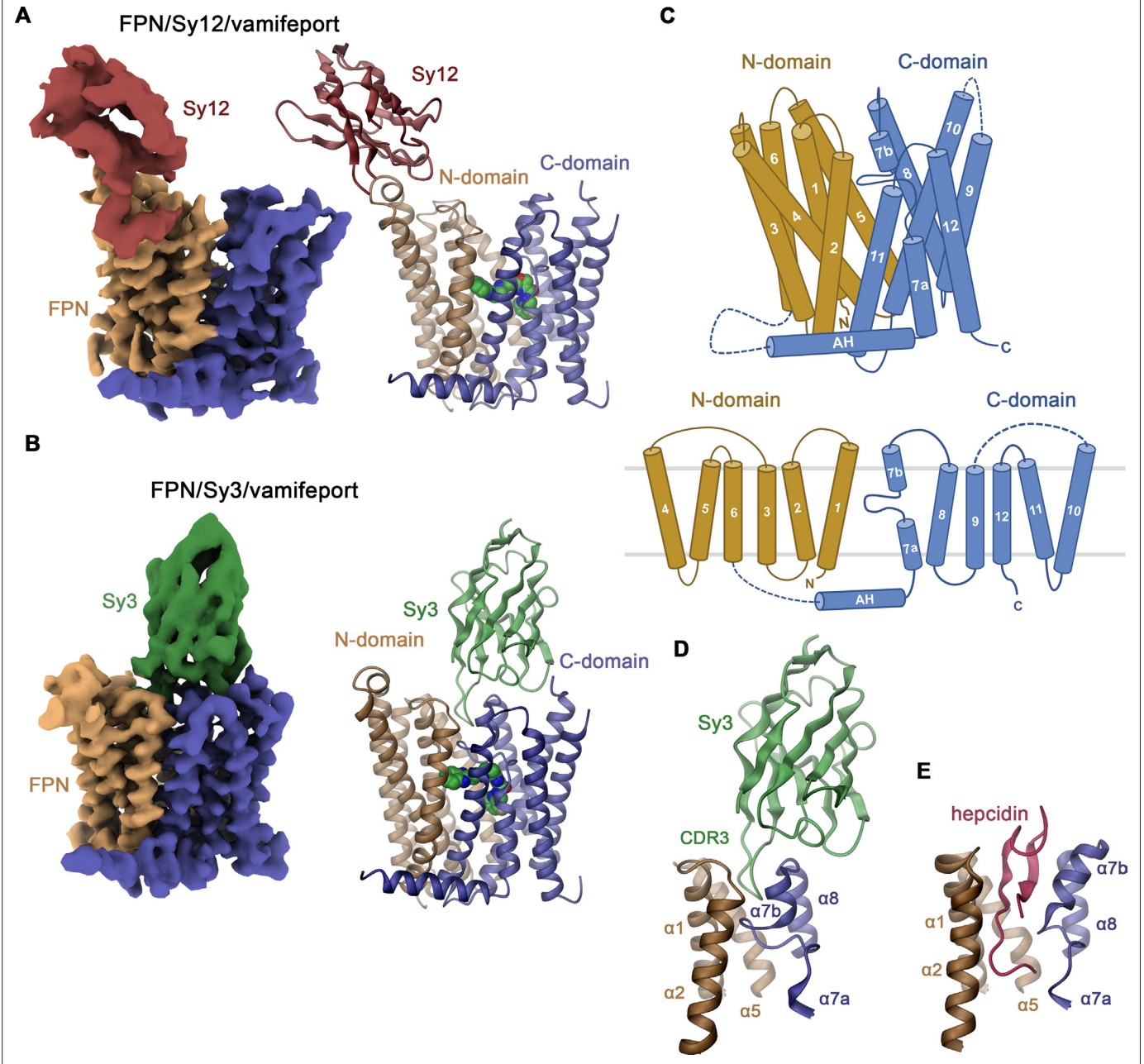

**Figure 2.** Structure of hsFPN/sybody complexes. Cryo-electron microscopy (cryo-EM) density (left) and ribbon model (right) of (**A**), the hsFPN/Sy12/ vamifeport and (**B**), the hsFPN/Sy3/vamifeport complex. N- and C-domains of hsFPN and the respective sybodies are labeled and shown in unique colors. Bound vamifeport is displayed in space-filling representation. (**C**) Schematic depiction of membrane-spanning helices (top) and topology of hsFPN (bottom). Ribbon representation of (**D**), the binding region of Sy3 and (**E**), hepcidin (obtained from PDBID: 6WBV). Secondary structure elements are labeled.

The online version of this article includes the following figure supplement(s) for figure 2:

**Figure supplement 1.** Cryo-electron microscopy (cryo-EM) reconstruction of the hsFPN/Sy12/vamifeport complex.

**Figure supplement 2.** Cryo-electron microscopy (cryo-EM) reconstruction of the hsFPN/Sy3/vamifeport complex.

**Figure supplement 3.** Cryo-electron microscopy (cryo-EM) reconstruction of the hsFPN/Sy3 complex.

**Figure supplement 4.** Cryo-electron microscopy (cryo-EM) density of hsFPN/sybody complexes.

**Figure supplement 5.** FPN sequence.

**Figure supplement 6.** hsFPN/sybody interactions.

loop 3 bridging the N- and C-domain, and the extracellular loop 5 connecting α9 and α10 (about 50 residues each) are not resolved. Although the cryo-EM density of Sy12 is insufficiently well defined for a detailed interpretation, it permitted the placement of a model consisting of the conserved part of the binder as rigid unit (*Figure 2A*, *Figure 2—figure supplement 4*). In this position, the interaction with the extracellular loop connecting α3 and α4 located at the periphery of the N-domain is apparent (*Figure 2A*, *Figure 2—figure supplement 6A*). Notably, the same loop is recognized by Fab45D8, which was used to determine hsFPN structures in a previous study (*Figure 2—figure supplement 6B*; *Billesbølle et al., 2020*). In contrast to the moderate quality of Sy12, the density of Sy3 in its complex with hsFPN is well resolved in both structures obtained in the absence and presence of vamifeport and has allowed the accurate interpretation of all three complementary determining regions (CDRs; *Figure 2B*, *Figure 2—figure supplements 2 and 4*). In its interaction with hsFPN, Sy3 buries an area of 2010 Å² of the combined molecular surface, with CDR-1 and CDR-2 binding to the periphery of α7, α8, and α10 and CDR-3 making extensive contacts with a pocket located between the two subdomains of the transporter (involving helices α1, α5, α7b, and α8; *Figure 2D*, *Figure 2—figure supplement 6C*). CDR-3 occupies a similar location as hepcidin, although the latter protrudes deeper into the pocket toward the center of the transporter, thereby establishing more extended contacts with α1 (*Figure 2E*).

In the Sy12 complex, hsFPN adopts a familiar outward-facing conformation (*Figure 3A*) that was previously observed in structures of the hsFPN-Fab45D8 (PDBID 6W4S) and tsFPN-Fab11F9 (PDBID 6VYH) complexes in the absence of bound hepcidin (with RMSDs of Cα positions of 0.939 Å and 0.944 Å, respectively, *Figure 3—figure supplement 1*; *Billesbølle et al., 2020*; *Pan et al., 2020*). Conversely, in both hsFPN/Sy3 complexes, α7b and the upper part of α8 have rearranged to approach the N-terminal domain by 3–4 Å relative to the hsFPN/Sy12/vamifeport complex, thus displaying a novel conformation of the transporter (*Figure 3B–D*). In this structure, residues on α7b (including Tyr 333) contact α1 and the connecting loop to α2 to increase interactions between both domains (*Figure 3D and E*). In addition, the lower part of the α7b helix unfolds with the extended α7a-b loop mediating additional contacts to α1 and α5 to occlude the access to the spacious aqueous pocket located in the center of the transporter (*Figure 3D and E*), whereas the intracellular part of the protein is unchanged (*Figure 3C*). Collectively, our structural data display hsFPN in two distinct conformations, an outward-open conformation in the hsFPN/Sy12/vamifeport complex and an occluded conformation in hsFPN/Sy3 complexes in the absence and presence of vamifeport. The latter are stabilized by the binding of Sy3 to a cleft at the interface between both domains (*Figure 2B and D*).

## Structural characterization of the vamifeport binding site

In the hsFPN/Sy3 map obtained from a sample containing vamifeport, we noticed a pronounced elongated density of appropriate size and shape of the inhibitor that is not present in a structure of the same protein complex obtained in its absence, where also the proximal α7a–α7b loop appears more mobile (*Figure 4A*, *Figure 4—figure supplement 1A–D*). Residual density at an equivalent location is also observed in the outward-facing structure of the hsFPN/Sy12 complex obtained in the presence of vamifeport (*Figure 4—figure supplement 1E, F*). However, the density is, in this case, less well defined, which likely reflects a preferred binding of the molecule to the occluded state, although it might in part also be a consequence of the lower resolution of the map and the compromised quality resulting from the described preferential orientation of particles.

The density attributed to vamifeport is located toward the center of FPN directly below the α7a–7b loop at the extracellular rim of a spacious occluded cavity where it is surrounded by residues on α2, α7a, α10, and α11 (*Figure 4B–D*). In this position, it overlaps with the binding site of hepcidin, which locks the transporter in its outward-facing state (*Figure 4—figure supplement 2*). The density, although strong throughout, is most pronounced at its ends defining the position of the two peripheral aromatic groups of the compound, which are both located in predominantly hydrophobic pockets (*Figure 4A and E*). The pocket located in the N-domain of hsFPN (N-pocket) involves the same residues that also contact the N-terminal part of hepcidin (i.e. Tyr 64, Val 68 on α2 and Tyr 501 and Phe 508 on α11; *Figure 4E and F*, *Figure 4—figure supplement 2B*). The connecting linker containing the aromatic oxazole ring is within 3–4 Å distance to the side chains of Asp 504 and His 507 on α11, Cys 326, Thr 329, and several backbone carbonyl groups on the α7a–7b loop and Arg 466 on α10, all of which are also involved in hepcidin binding (*Figure 4E–G*, *Figure 4—figure supplement 2B–D*).

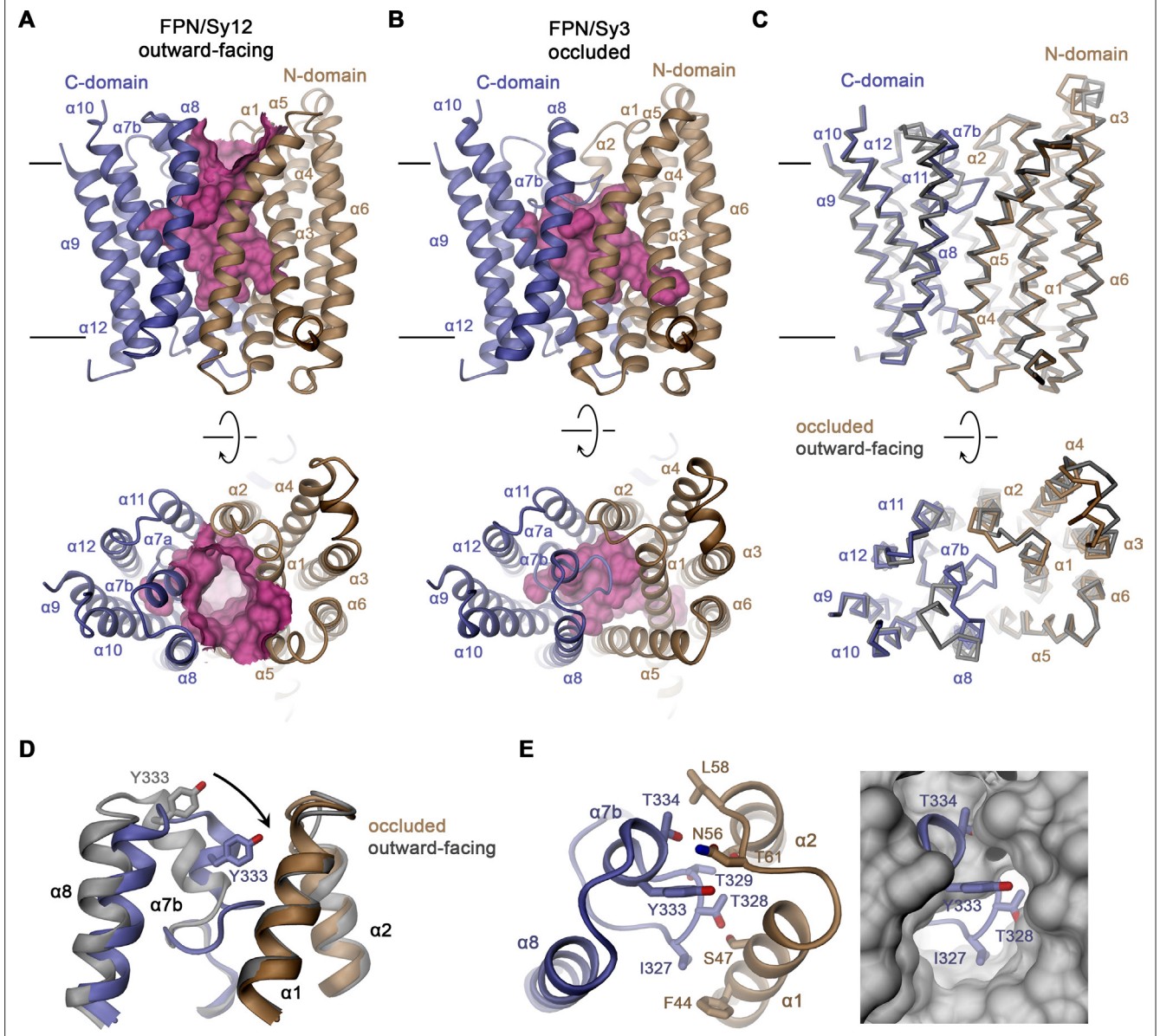

**Figure 3.** Features of hsFPN conformations. (**A**) Outward-facing conformation of hsFPN as observed in the hsFPN/Sy12/vamifeport complex. (**B**) Occluded conformation of hsFPN as observed in the hsFPN/Sy3/vamifeport complex. (**A and B**) The molecular surface of the outward-facing and occluded cavities harboring the substrate binding sites is shown in magenta. (**C**) Cα representation of a superposition of both conformations of hsFPN. (**A–C**) Views are from within the membrane (top) and from the outside (bottom). (**D**) Comparison of both conformations in the region showing the largest differences. (**E**) Interaction between α7b and α2 and α3 closing the extracellular access to the substrate binding site (left). To further illustrate the occlusion by residues of the α7a–α7b region, the occluded model is superimposed on a surface representation of the outward-facing structure (right). (**C and D**) The outward-facing structure is colored in gray for comparison. (**A–E**) Secondary structure elements are labeled. The coloring is as in *Figure 1A*.

The online version of this article includes the following figure supplement(s) for figure 3:

**Figure supplement 1.** Superposition of outward-facing conformations.

In contrast, the pocket in the C-terminal domain (C-pocket) is surrounded by residues most of which are not involved in hepcidin binding, including Leu 314, Leu 317, Tyr 318, and Thr 320 on α7a and Leu 469, Trp 470 and Asp 473 on α10 (*Figure 4E and G*).

Vamifeport can also be modeled into this density in an alternative orientation, with the terminal fluoro-pyridine and benzimidazole groups located in the opposite pockets (*Figure 4—figure supplement 3A, B*). Between both possible binding modes of the compound, the initially described one with the benzimidazole group in the N-pocket and the fluoro-pyridine group in the C-pocket allows

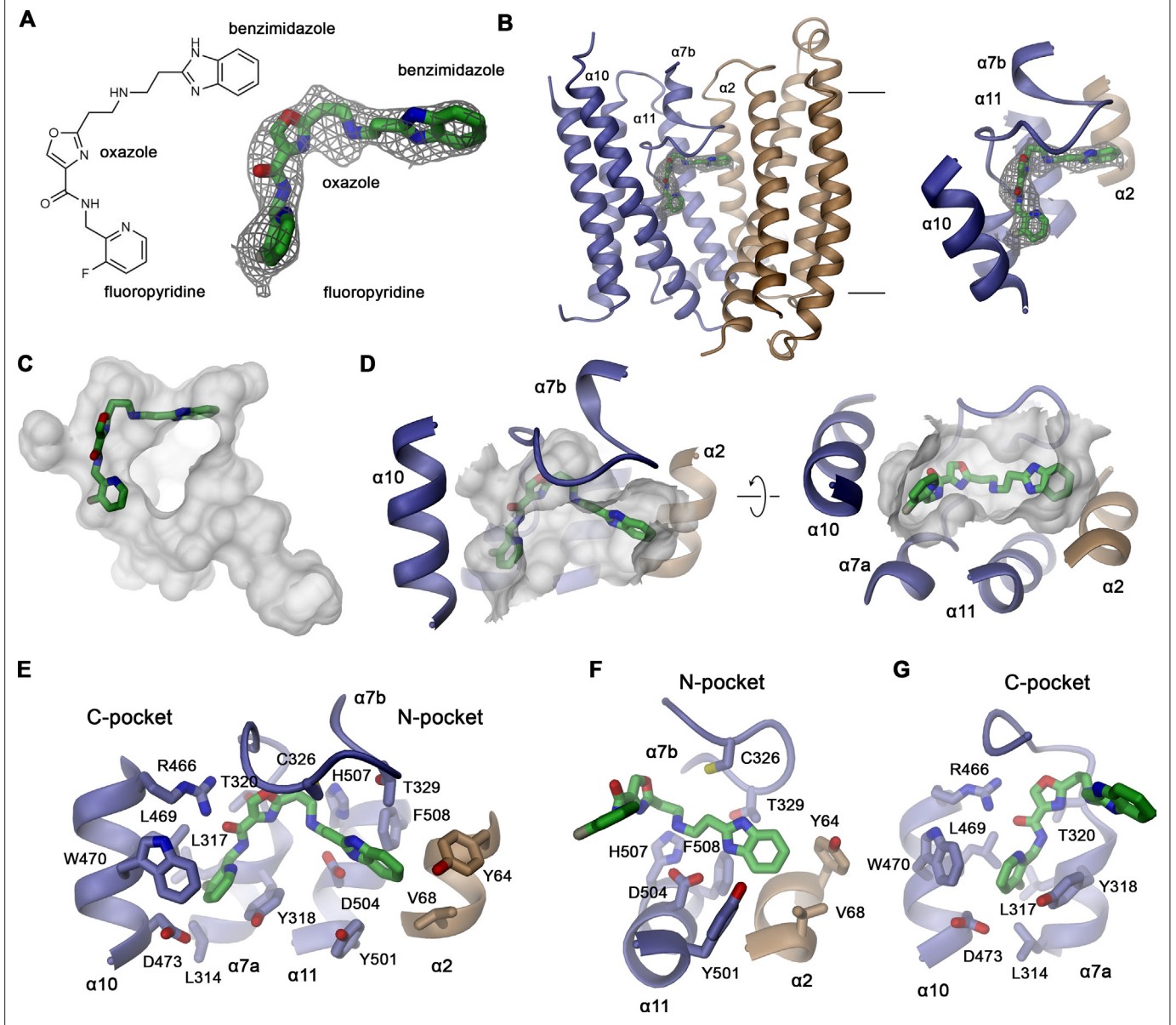

**Figure 4.** Structural basis of vamifeport binding. (**A**) Chemical structure of vamifeport and its fit into the corresponding cryo-electron microscopy (cryo-EM) density of the hsFPN/Sy3/vamifeport complex. (**B**) Ribbon representation of the occluded conformation of FPN observed in the hsFPN/Sy3/vamifeport complex with the cryo-EM density attributed to vamifeport shown to illustrate its location within the transporter. Inset (right) shows blow-up of the binding region. (**A and B**) Cryo-EM density is contoured at 6.5 σ. (**C**) Location of vamifeport within the occluded cavity of hsFPN and (**D**), blowup of the binding pocket. (**D and E**) The molecular surface of the cavity is shown in gray. (**E**) Protein inhibitor interactions in the vamifeport binding site and interactions in (**F**) the N-pocket and (**G**) the C-pocket. (**E–G**) Side chains of interacting residues are shown as sticks and labeled.

The online version of this article includes the following source data and figure supplement(s) for figure 4:

**Figure supplement 1.** Density in the vamifeport binding site.

**Figure supplement 2.** Hepcidin interactions.

**Figure supplement 3.** Alternative vamifeport binding mode.

**Figure supplement 4.** Evaluation of vamifeport interactions.

**Figure supplement 4—source data 1.** Total energies of hsFPN-vamifeport complex in original and alternative conformation obtained by docking calculations and CAMPARI keyword files and input and output files for docking calculations.

**Figure supplement 5.** Interactions in the metal ion binding sites.

for a larger number of interactions with the protein (*Figure 4E*, *Figure 4—figure supplement 3B–F*). In this mode, the carbonyl of the amide between the fluoro-pyridine and the oxazole group would interact with the side chain of Arg 466, and the oxazole group would be placed in a region with more pronounced density in contact with Thr 320 (*Figure 4—figure supplement 3C, D*). The secondary amine group on the linker, which is likely protonated, and one of the nitrogen atoms on the benzimidazole group would reside in proximity to Asp 504, whereas the other nitrogen in the heterocycle would contact Cys 326 and Thr 329 (*Figure 4—figure supplement 3C, D*). In the alternative binding mode of vamifeport, the benzimidazole group would bind to the C-pocket with one of its nitrogen atoms and the secondary amine of the linker located in proximity to the Arg 466 side chain leading to a potential electrostatic repulsion, which renders this interaction less favorable (*Figure 4—figure supplement 3E, F*). The preferred interactions in the initially described orientation of vamifeport are also reflected in the 12–20 kcal/mol lower docking energy of the compound, which is dominated by the more favorable polar interactions established with the protein (*Figure 4—figure supplement 4*).

Besides overlapping with the hepcidin binding site, vamifeport is located in close proximity to the S2 metal binding site with His 507 on α11 and Cys 326 on the α7a–α7b loop acting as coordinating residues (*Figure 4F*, *Figure 4—figure supplement 5A*). In all present mammalian FPN structures, the position of His 507 displays only small variations, whereas the position of Cys 326 on the flexible loop shows large differences (*Figure 4—figure supplement 5A, B*). In the presence of bound metal ions, His 507 and Cys 326 are located about 3.5 Å from each other, while in the same outward-facing conformation in the absence of substrates, the separation increases to 6 Å. In case of both hsFPN/Sy3 structures, Cys 326 moves even further toward the N-terminal domain and extends the distance to His 507 to 7–9 Å, which likely prohibits the concomitant binding of vamifeport and a divalent metal ion at the S2 site (*Figure 4—figure supplement 5A, B*).

## Functional characterization of the vamifeport binding site

To validate the binding mode of vamifeport, we have mutated residues of the protein that are in contact with the compound and employed our TMR-hepcidin displacement assay to determine whether these mutants would weaken binding to hsFPN. To this end, we have introduced point mutations in 10 amino acids lining the binding site (i.e. Tyr 64, Val 68, Tyr 501, Phe 508, Leu 314, Leu 317, Tyr 318, Leu 469, Trp 470, and Asp 473). These residues were mutated to alanine and serine to either truncate the side chain or introduce a polar moiety. In most cases, the expression levels of respective mutants were reduced compared to wild type (WT), and in case of Tyr 318, neither mutant was expressed (*Figure 5—figure supplement 1*). In addition, we have mutated Arg 466 and Asp 504 to alanine, with the former being well expressed and the latter at about one third of the WT level (*Figure 5—figure supplement 1*).

For displacement assays, we selected V68S and Y501S of the N-pocket, L469A, L469S, and W470S of the C-pocket, and R466A and D504A contacting the linker region. We first determined whether these mutants would still bind TMR-hepcidin by titrating the tagged peptide hormone and monitoring its FP (*Figure 5—figure supplement 2*). In these initial experiments, we found mutations of

**Table 2.** Hepcidin and vamifeport binding.

| hsFPN construct | $K_D$ TMR-hepcidin (nM) | $K_D$ hepcidin-25 (nM) | $K_D$ vamifeport (nM) |
|---|---|---|---|
| WT | 100±4 | 131±12 | 24±3 |
| R466A | 501±22 | 713±117 | 83±16 |
| L469A | 57±2 | 108±10 | 37±4 |
| L469S | 94±3 | 180±17 | 60±7 |
| W470S | 227±8 | 222±19 | 82±11 |
| V68S | 1044±60 | n.d. | n.d. |
| Y501S | >>1000 | n.d. | n.d. |
| D504A | >>1000 | n.d. | n.d. |

n.d., not determined.

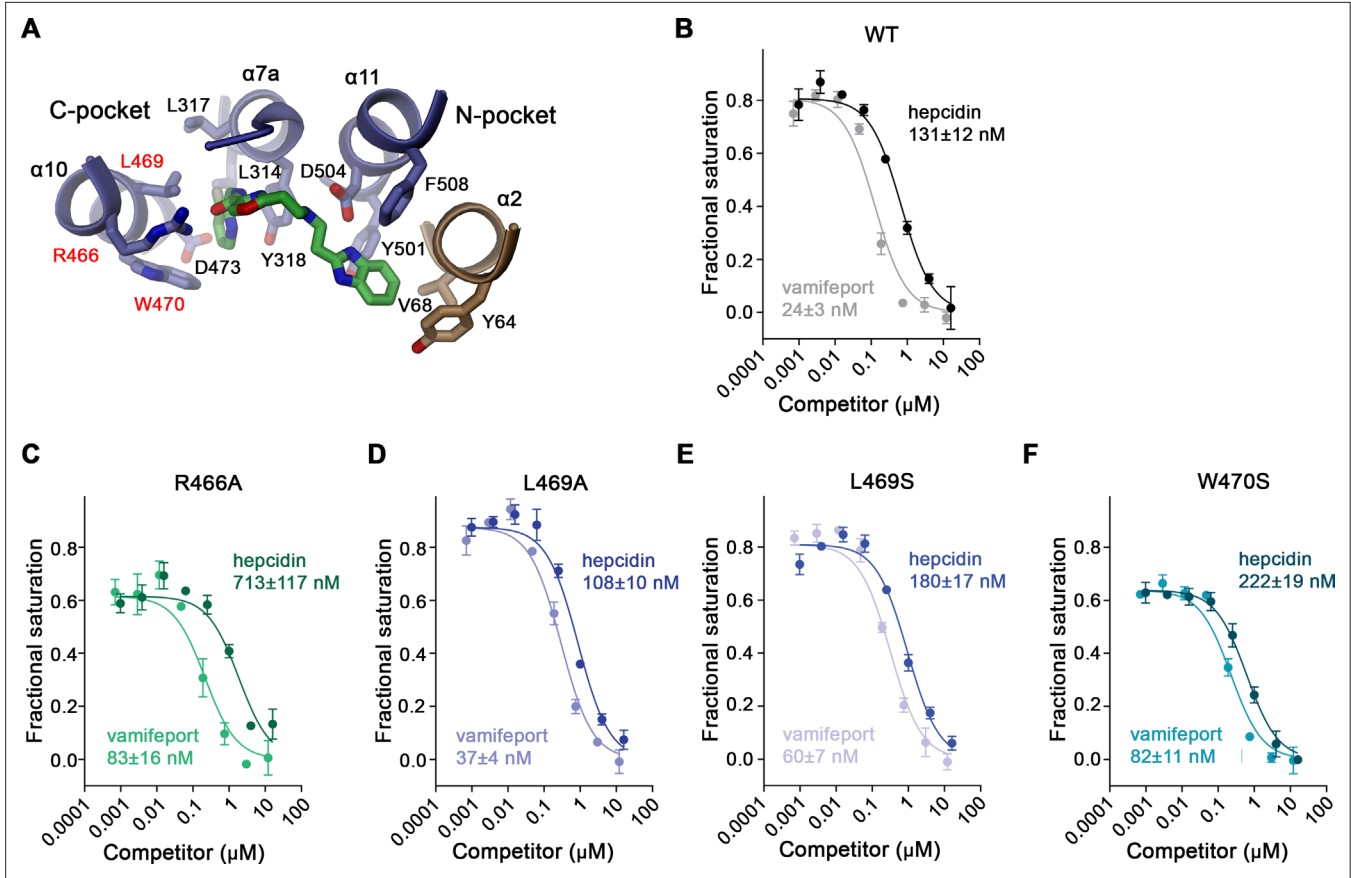

**Figure 5.** Properties of binding site mutants. (**A**) Close-up of vamifeport/FPN interactions with mutated residues. Residues of the C-pocket whose mutations were characterized in detail are labeled in red. Displacement of TMR-hepcidin with unlabeled hepcidin-25 or vamifeport measured by changes in the fluorescence polarization of labeled hepcidin. Investigated constructs show: (**B**) hsFPN WT and the point mutants, (**C**) R466A, (**D**) L469A, (**E**) L469S, and (**F**) W470S. The $K_D$ values for the interaction between FPN/hepcidin-25 and FPN/vamifeport are displayed. Values were obtained from a fit to a competition binding model (lines) described in the methods. (**B–F**) Data show mean of three independent measurements; errors are SEM. Similar results were obtained in independent experiments using different preparations and concentrations of hsFPN.

The online version of this article includes the following source data and figure supplement(s) for figure 5:

**Source data 1.** TMR-hepcidin fluorescence polarization displacement data of hsFPN WT and point mutants.

**Figure supplement 1.** Expression of hsFPN mutants.

**Figure supplement 1—source data 1.** Fluorescence detection size-exclusion chromatography data of detergent solubilized extracts of hsFPN WT and point mutants harboring a C-terminal GFP-tag.

**Figure supplement 2.** Hepcidin binding to hsFPN mutants.

**Figure supplement 2—source data 1.** TMR-hepcidin fluorescence polarization direct binding data of hsFPN WT and point mutants.

the N-pocket as well as Asp 504 and Arg 466 to drastically reduce the affinity to TMR-hepcidin, which is expected in light of their contribution to hepcidin binding (*Figure 4—figure supplement 2B, D*). Specifically, TMR-hepcidin interacts with mutants Y501S and D504A with μM affinity, with curves overlapping with a WT titration performed in presence of a high excess of vamifeport to block specific TMR-hepcidin binding, suggesting that TMR-hepcidin binding to these mutants is non-specific (*Figure 5—figure supplement 2A*). V68S bound TMR-hepcidin with considerably lower affinity ($K_D$ of 1044 nM vs 100 nM for WT) and for R466A, we determined a $K_D$ value of 501 nM (*Figure 5—figure supplement 2A*). In contrast, mutants of the C-pocket bound TMR-hepcidin with similar affinity as WT (i.e. L469A 57 nM; L469S 94 nM; W470S 227 nM; *Figure 5—figure supplement 2B*, *Table 2*). Subsequent experiments performed for mutants of the C-pocket and R466A showed displacement of TMR-hepcidin by hepcidin-25, underlining the specificity of its binding (*Figure 5B*, *Table 2*). As expected, in these competition assays, R466A displayed an elevated $K_D$ (713 nM vs 130 nM for WT), while the

values for L469S and W470S were in the same range as for WT (108 nM for L469A, 180 nM for L469S, and 222 nM for W470S; *Figure 5C–F*, *Table 2*). Similarly, in all mutants, vamifeport did still compete with TMR-hepcidin, however, with considerably lower affinity than WT (25 nM; *Figure 5B*, *Table 2*). Specifically, the fit for R466A resulted in a $K_D$ of 83 nM, and truncation of the side chains comprising the hydrophobic C-pocket resulted in $K_D$ values of 37 nM (L469A), 60 nM (L469S), and 82 nM (W470S), implying that these residues indeed contribute to the binding of vamifeport (*Figure 5C–F*, *Table 2*), thus further supporting our structural results defining the vamifeport binding site.

## Discussion

Here, we have investigated the interaction between human FPN (hsFPN) and its inhibitor vamifeport, which was shown to ameliorate anemia and iron homeostasis in a mouse model of β-thalassemia and is now in clinical development for this disorder and sickle cell disease (*Manolova et al., 2019*; *Nyffenegger et al., 2022*). Our study has provided two major novel results, it has revealed the structure of an occluded state of hsFPN, where the access to the substrate binding sites is blocked from both sides of the membrane (*Figure 3*), and it has defined the interactions with vamifeport in the same occluded conformation (*Figure 4*).

The structures determined in the course of our investigations were obtained using two sybodies binding to different epitopes of hsFPN, which both stabilize distinct conformations of the transporter (*Figure 2*). Whereas, the hsFPN/Sy12/vamifeport complex shows an outward-facing conformation in the absence of bound metal ions that closely resembles equivalent states determined in previous studies (*Billesbølle et al., 2020*; *Pan et al., 2020*), two hsFPN/Sy3 structures obtained in the absence and presence of vamifeport display a previously unknown conformation of FPN, where the access to substrate binding sites from the outside has closed, while the intracellular part of the transporter has remained unchanged (*Figure 3*). In both structures in complex with Sy3, helix α7b, located at the extracellular part of the C-domain, has moved toward the N-domain while unwinding its N-terminal turn. The resulting extended loop connecting α7a with α7b bridges toward α-helices 1 and 2 to seal off the outward-facing cavity leading to the substrate binding sites (*Figure 3C–E*). Since this conformation with occluded substrate binding pocket is stabilized by the binding of Sy3 from the extracellular side to a cleft between both sub-domains involving α-helices 1, 5, 7, and 8 (*Figure 2D*), it is currently not known to which degree it represents an intermediate in the transport cycle. Notwithstanding the ambiguity concerning its functional relevance, several lines of evidence suggest that this conformation would be assumed in solution and at least be close to conformations on the transition toward the inward-facing state. In the sybody selection process, binders frequently target pre-existing conformations or ones that are energetically close. Consequently, we find it unlikely that this conformation would be solely induced by sybody binding, particularly since it is also targeted by vamifeport, which was not present during the selection process. Moreover, the observed occluded conformation bears features that directly relate to the differences between distinct states of known FPN structures. The mobility of α7b and α2 is evident in the comparison between outward-facing structures of FPN obtained in the absence and presence of hepcidin, where binding of the latter causes a further opening of the aqueous cleft to accommodate the bound peptide hormone (*Billesbølle et al., 2020*; *Figure 6A*). Conversely, the same structural elements of the occluded conformation are positioned between the outward- and inward-facing structures of the close bacterial homolog bbFPN (*Taniguchi et al., 2015*; *Figure 6B*). Finally, the closure of the extracellular entrance by the movement of α7b toward the N-terminal domain is also found in other members of the MFS that transport sugars and peptides (*Dang et al., 2010*; *Huang et al., 2003*; *Newstead et al., 2011*). This is best illustrated in transporters of the GLUT family on their transition between outward and occluded states, where this helix was assigned as extracellular gating element (*Deng et al., 2015*; *Drew and Boudker, 2016*; *Drew et al., 2021*; *Quistgaard et al., 2016*; *Figure 6—figure supplement 1A, B*). Compared to other transporters sharing the MFS fold, however, the unwound part of α7 in the occluded conformation of FPN is unusually long. While somewhat flexible in the same conformation obtained in the absence of vamifeport, it rigidifies in its presence as a consequence of inhibitor interactions, which presumably further stabilize this state (*Figure 4—figure supplement 1A–D*). Hence, although weak cryo-EM density indicates binding of the inhibitor to a similar position also in the outward-facing conformation of the hsFPN/Sy12/vamifeport complex, the much better defined density implies stronger interactions

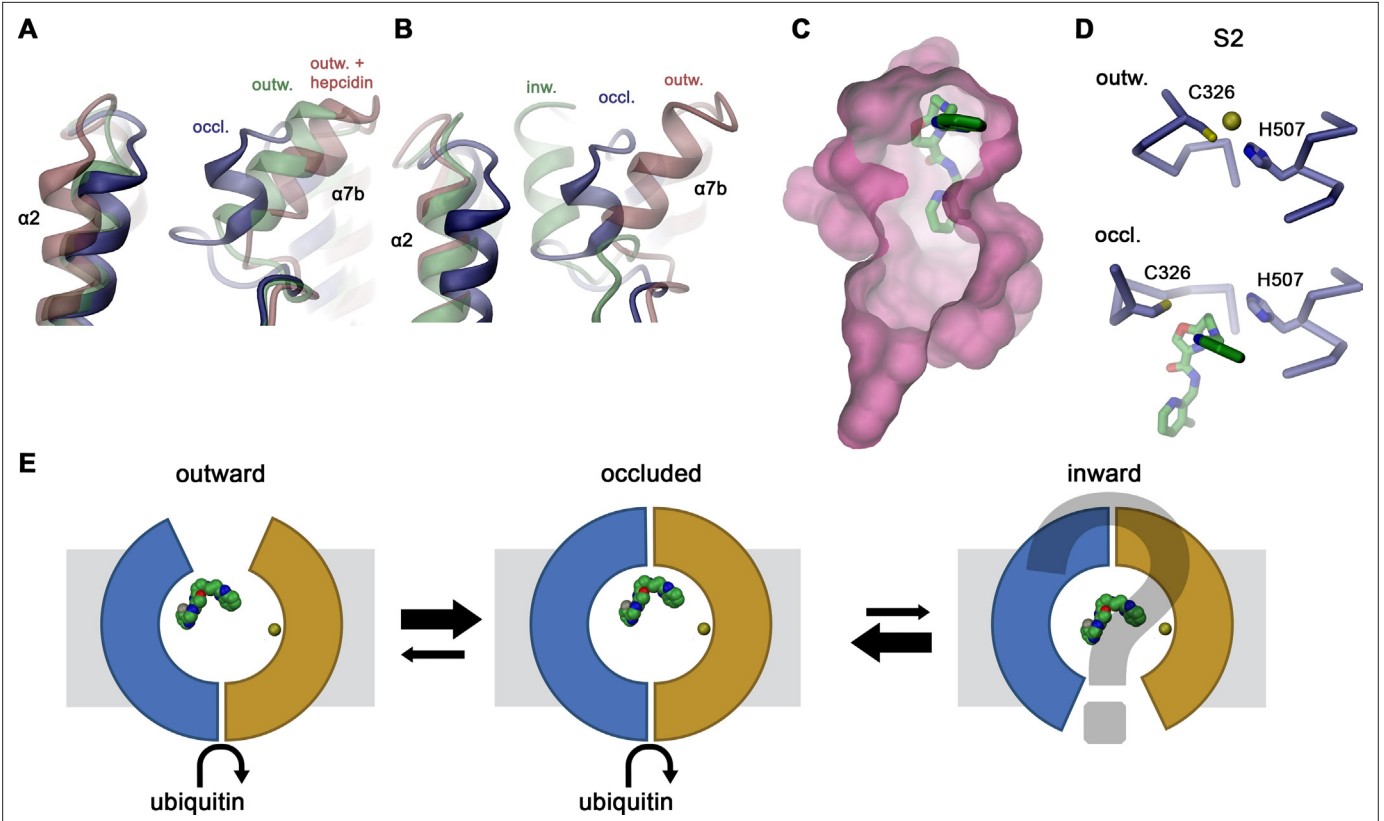

**Figure 6.** Hypothetical mechanism of FPN regulation by vamifeport. Conformational plasticity of the extracellular region surrounding helix α7b as illustrated by a superposition of the occluded conformation of hsFPN with (**A**) hsFPN in outward-facing apo (PDBID 6W4S) and hepcidin-bound states (PDBID 6WBV) and (**B**) the prokaryotic bbFPN in outward- and inward-facing states (PDBID 5AYN and 5AYO). (**C**) Location of vamifeport at the extracellular rim of the occluded cavity leaves sufficient space for larger inhibitors. (**D**) Comparison of the metal ion binding site S2 in the outward-facing conformation of hsFPN (PDBID 6VYH) and its occluded conformation indicates competition of vamifeport with the ion binding to S2. (**E**) Hypothetical relationship between vamifeport-bound FPN conformations and the ability of the protein to get ubiquitinated. The possible binding of $Fe^{2+}$ to the S1 site, which is not overlapping with the vamifeport binding site, is indicated. The stabilization of the occluded conformation might potentially facilitate the recruitment of the ubiquitination machinery on the intracellular side.

The online version of this article includes the following figure supplement(s) for figure 6:

**Figure supplement 1.** Conformational changes and substrate binding in different transporters.

in the occluded conformation, suggesting that the latter would be preferentially adopted in complex with the compound in a cellular environment (***Figure 4A***, ***Figure 4—figure supplement 1A, E***).

The spacious protein-enclosed aqueous region that is formed in the occluded conformation of hsFPN encompasses a volume of 906 $Å^3$, which is about three times larger than the volume of vamifeport (***Figure 6C***). By binding at its extracellular rim, vamifeport forms extended tight interactions with the protein at one of its faces, whereas the opposite face of the inhibitor is exposed to the trapped aqueous environment of the cavern, which might accommodate even considerably larger molecules. In this location, the terminal benzimidazole and fluoro-pyridine groups are embedded in hydrophobic pockets, with the N-pocket being constituted by residues on α2 and α11 and the C-pocket by residues on α7a and α10, whereas the linker containing an oxazole ring and connecting both aromatic groups is contacted by residues on α10, 11, and the α7a–7b loop (***Figure 4E–G***). These residues are highly conserved throughout FPN orthologs, and they are identical in all mammalian family members (***Figure 2—figure supplement 5B***), explaining the inhibitory action of vamifeport in mice (***Manolova et al., 2019***). The involvement of the N-pocket and the region contacting the vamifeport linker in the binding of hepcidin underlies the competitive relationship between both ligands despite their stabilization of distinct protein conformations (***Figure 4—figure supplement 2B, C***). In contrast, the observed interactions in the C-pocket are unique to vamifeport, with mutations in this site lowering its affinity to hsFPN, consistent with our structural data (***Figure 5C–F***).

Notwithstanding the unknown functional correspondence of the described occluded conformation, we expect a spacious aqueous cavity to be retained during transport by FPN, which is uncommon for selective metal ion transporters, where bound substrates are usually tightly surrounded by protein residues (*Figure 6—figure supplement 1C–F*). In contrast, the two substrates binding sites (S1 and S2) identified in the outward-facing state of FPN are located at two disconnected locations toward the extracellular border of the cavern (*Billesbølle et al., 2020*; *Pan et al., 2020*; *Figure 6—figure supplement 1C, D*). Both sites offer only few protein interactions, with surrounding water molecules likely completing the preferred octahedral metal ion coordination geometry (*Figure 6—figure supplement 1E*). The organization of these sites is in striking contrast to unrelated transition metal ion transporters of the SLC11 family, which tightly coordinate their substrate by protein residues in narrow pockets that exclude most of the solvent (*Bozzi et al., 2019*; *Ehrnstorfer et al., 2014*; *Ray et al., 2022*; *Figure 6—figure supplement 1F*), thus emphasizing the distinct transport mechanism of FPN that is still not fully understood.

In the inhibition of transport, a presumable interference between vamifeport and transported metal ion binding appears plausible. The overlap of the bound inhibitor with the S2 site suggests a competition with metal ion binding at this location, whereas the S1 site would not be affected, although experimental evidence for such competition still needs to be provided (*Figure 6D*, *Figure 4—figure supplement 5*). Similarly, the primary mechanism of how vamifeport reduces iron transport has remained ambiguous as it is not clear whether this would be accomplished by a slowdown or block of transport by locking the transporter in a single conformation or by its displacement from the plasma membrane. As previously shown, vamifeport and hepcidin affect hsFPN in a similar way, both leading to ubiquitination, internalization, and degradation of the transporter (*Manolova et al., 2019*). However, compared to hepcidin, the effect of vamifeport is less pronounced, resulting in a mere slow-down of the process. Thus, the question remains how these extracellular signals are relayed to the intracellular side to promote ubiquitination. Binding of hepcidin locks the transporter in an outward-facing state with the peptide hormone impeding structural transitions (*Billesbølle et al., 2020*; *Pan et al., 2020*; *Figure 4—figure supplement 2A*). This locked state might facilitate the binding of ubiquitin ligases on the intracellular side. In contrast, our data suggest that the binding of the smaller ligand vamifeport would stabilize an occluded state, which shares its intracellular conformation with the outward-facing state (*Figures 3C and 6E*) and might thus exert a comparable effect. The transition from an outward facing to an occluded state raises the question of whether the transporter could further transit into an inward-facing state even in the presence of vamifeport, which appears possible considering the spacious cavity found in the inward-facing state of a bacterial FPN homolog, which could accommodate a molecule with the size of vamifeport (*Taniguchi et al., 2015*). It is thus conceivable that, while stabilizing an intermediate state, the binding of vamifeport does not completely prohibit the transition of FPN into an inward-facing conformation (*Figure 6E*). The resulting less stringent impediment of movement compared to hepcidin might explain the altered phenotype of vamifeport with respect to ubiquitination, internalization, and degradation of FPN, although the detailed mechanism of inhibition still needs to be elucidated (*Manolova et al., 2019*). In combination, our work has provided novel insight into the conformational properties of a central regulator of iron homeostasis in humans and defined its interaction with a compound of high therapeutic relevance. It does thus provide the basis for the development of novel classes of molecules with improved pharmacological properties.

## Materials and methods
### Construct preparation
The human FPN (hsFPN, UniProt identifier Q9NP59) gene was codon optimized for expression in a mammalian expression system (*Table 3*), synthesized by GeneScript and cloned by FX-cloning (*Geertsma and Dutzler, 2011*) into the expression vector pcDX3cGMS, which adds an HRV 3C protease cleavage site, GFP, a myc-tag and streptactin-binding peptide to the C-terminus of the protein. Sybody sequences (*Table 3*) were cloned into the pBSinit vector (*Zimmermann et al., 2020*), an FX-compatible, chloramphenicol resistant, arabinose-inducible vector containing an N-terminal pelB leader sequence and a C-terminal His$_6$ tag. Point mutations were introduced by site-directed mutagenesis (*Table 4*; *Li et al., 2008*).

**Table 3.** DNA sequences.

**Codon optimized sequence of hsFPN**

GTACCAGAGCCGGAGATCACAACGGCAGAGAGGCTGCTGCGGCAGCCTGGCAGATTACCTGACCTGTACCTGGGCCATTCTGCTGACGCACATGGGGCGAC
AGAATGTGGCACTTCGCGCGTGTCCGTCTTTCTCGTGGAACTGTATGGAAACAGCCTGCTGCTGACTGGTGGTCCGGAGCCATTATA
GGCGATTGGGTCGACAAGAAACGCCCGGCTGAAAGTGGCCAAACAAGCCTGGTGGTGCAGAAACGTGAGCGTGATCATCGTGTGCATGCCCAGCACCGCTACCGCCATCACCATTCAAAGAGATTG
CGAACTGCTGACAATGTACCACGGCTGGGTGCTGACCTCTTGCTACATCGTGATCATCACAATGAACCACCATCCTGGCTCCTATGGCCGTGGCCAGATCATGA
GATCGTGGTTGTGGCCGGCGACCGGAGGACCGGAGCAAGCTGGCTAACATGAACGCCACCATCCTGGCTCATCAGCGGCTTCATCAGCGGCTT
AAGGCCGGCCACGAGGAGGAAGGAGAGCGAGAGCAGCTGAAGCAGCTAAGCCTCGACAGATGGCCAGGCCCACCTGCGCCTTCAGATGGCCAGGCCTTTCCGGAGACAGCCGTGGGTGTCCTACTACAACCAGCCTAAGGCGCCGCCATGG
CGAACTGGAACACGAGCCAGGAGGAGAGCCCCACCTGCGCCCTCTCAGATGGCCGAGCCTTTCGGAGTGTCCTACTACAACCAGCCTGTGTTCCTGGCCGCCGGCATGG
GCCTGGCCTTCCTGTACAGTGGCTTTCACCTGGCTGGGATTCGATTGTATCACCACCGGCGACTTGTTAGAACCGGCCTGATCAGCGGCATCATCCAGGGCCTGATCAGCGGTCTTTA
GCATTATGGGTACAGTGGCTTTCACCTGTCTGTGTTCCCCATTCGAGGACATCAAGATTCATCCCGGAGATTACCACCGAAATCTACACATGA
TGCCTGGCAGCCCTCTGACCGTCGTCTGTTCGCCCTGAACATCGTGCCCATCATCTCTGTTCGCTGGATGGCCTGTGTGTCCTTTG
CAACGGGCTCCAACAGCGCTAACATCGTGCCGGAGACAAGCCCTGCTCCTGTTGCTGCCATCATCGGCCTGTGGTCATCGGC
ACCTGACACTGACCCAGCTGCTGCAGGGAACGTGATCGAGGAGCGAGAGCGGAGGAATCATCAACGCATGAACCTGCAGAACTACCTGCTGCACTTCATCATGGTCATCC
TAGCTCCAAACCCCGAGACCTTCGGCCTGCTCCTGATCAGCGTGAGCTTCGTGGCCACATACACTGGCCAACAAGCTGTTCGCAT
GCGGACCTGACGCCCAAGGAAGGTGCGGAAGGAAAATCAGGCCAATACCAGCGTGGTG

**Sequence of Sybody 1**

GTTCAGCTCGTTGAGAGCCGGTGGTGCGGCCTGGTCCAAGCTGGCGGTTCGCTCGCTCTGAGCTGCGCCGCAAGCCGGTTTCCCGGTGGAACAGGGTTGGATGGCTTGGTATCGTCAGGC
ACCGGGCCAAAGAACGTGAGTGGGTCGCGGCGATTTCTAGCTCGTTGGCATACGTTACCAAATCAGCCGCGACAAGCGGAAGAATACGGTCTA
TTTGCAGATGAATAGCCTGAAACCGGAAGATACCGCGGTTACTACTGTAAAGACAACACGTTGGTACTCTGCTCAGTAGCGACTATTGGGGCCAAGGTACCCAAGTGACTGTGAG
CGCAGGAAGAGCTGGCGAACAAAAACTCATCTCAGAAGAGAGGATCTGAATAGCGCCGTCGAC

**Sequence of Sybody 3**

GTTCAGCTCGTTGAGAGCCGGTGGTGCGGCCTGGTCCAAGCTGGCGGTTCGCTCGCTCTGAGCTGCGCCGCAAGCCGGTTTCCCGGTGGTTGGAACGAAATGCGTTGGTATCGTCAGGC
ACCGGGCCAAAGAACGTGAGTGGGTCGCGGCGATTCTGTTACTACGCAGATTCTGTTAAGGGCCGCTTTACCATCAGCCGCGACAACGCGAAGAATACGGTCTA
TTTGCAGATGAATAGCCTGAAACCGGAAGATACCGCGGTTTACTACGTGAAAGACTACGGTATGGCTTTCTGGTACTACGACTATTGGGGCCAAGGTACCCAAGTGACTGTGAG
CGCAGGAAGAGCTGGCGAACAAAAACTCATCTCAGAAGAGAGGATCTGAATAGCGCCGTCGAC

**Sequence of Sybody 8**

GTTCAGCTCGTTGAGAGCCGGTGGTGCGGCCTGGTCCAAGCTGGCGGTTCGCTCGCTCTGAGCTGCGCCGCAAGCCGGTTTCCCGGTGTACGCTTACAACATGGAATGGTATCGTCAGGC
ACCGGGCCAAAGAACGTGAGTGGGTCGCGGCGATTGCTAGCTACGCTAGCTACGCAGATTCTGTTAAGGGCCGCTTACCATCAGCCGCGACAACGCGAAGAATACGGTCTA
TTTGCAGATGAATAGCCTGAAACCGGAAGATACCGCGGTTTACTACGTGAAAGACTACGGTTACCTGTGGTACCATTACGACTATTGGGGCCAAGGTACCCAAGTGACTGTGAG
CGCAGGAAGAGCTGGCGAACAAAAACTCATCTCAGAAGAGAGGATCTGAATAGCGCCGTCGAC

**Sequence of Sybody 11**

GTTCAGCTCGTTGAGAGCCGGTGGTGCGGCCTGGTCCAAGCTGGCGGTTCGCTCGCTCTGAGCTGCGCCGCAAGCCGGTTTCCCGGTGTACTCTCGTGGTATGTACTGGTATCGTCAGGC
ACCGGGCCAAAGAACGTGAGTGGGTCGCGGCGATTCATAGCTACGCAGATTCTGTTAAGGGCCGCTTTACCATCAGCCGCGACAACGCGAAGAATACGGTCTA
TTTGCAGATGAATAGCCTGAAACCGGAAGATACCGCGGTTTACTACTGTAAACGTGAAAGACATGGGTTACCTGGTTACCAGTACGACTATTGGGGCCAAGGTACCCAAGTGACTGTGAG
CGCAGGAAGAGCTGGCGAACAAAAACTCATCTCAGAAGAGAGGATCTGAATAGCGCCGTCGAC

**Sequence of Sybody 12**

GTTCAGCTCGTTGAGAGCCGGTGGTGCGGCCTGGTCCAAGCTGGCGGTTCGCTCGCTCTGAGCTGCGCCGCAAGCCGGTTTCCCGGTGTGGAACTCTGTATGTTCGGTATCGTCAGGC
ACCGGGCCAAAGAACGTGAGTGGGTCGCGGCGAAAGGTTACTCTACGCATTACGCAGATTCTGTTAAGGGCCGCTTTACCATCAGCCGCGACAACGCGAAGAATACGGTCTA
TTTGCAGATGAATAGCCTGAAACCGGAAGATACCGCGGTTTACTACTGTAAACGTGAAAGACGACTTTGGTGGTGCAGACTATTGGGGCCAAGGTACCCAAGTGACTGTGAG
CGCAGGAAGAGCTGGCGAACAAAAACTCATCTCAGAAGAGAGGATCTGAATAGCGCCGTCGAC

**Table 4.** Primers for mutagenesis.

| Mutation | Primer sequences |
| --- | --- |
| Y64A | for: GCTGACTGCTGTTGCCGGCCTGGTGGTGGCCGGCAGC<br>rev: AACAGCAGTCAGCAGCAGGCTGTTTCCATACAGTTCCACG |
| Y64S | for: GCTGACTGCTGTTTCCGGCCTGGTGGTGGCCGGCAGC<br>rev: AACAGCAGTCAGCAGCAGGCTGTTTCCATACAGTTCCACG |
| V68A | for: GCTGACTGCTGTTTACGGCCTGGTGGCGGCCGGCAGC<br>rev: AACAGCAGTCAGCAGCAGGCTGTTTCCATACAGTTCCACG |
| Y68S | for: GCTGACTGCTGTTTACGGCCTGGTGTCGGCCGGCAGC<br>rev: AACAGCAGTCAGCAGCAGGCTGTTTCCATACAGTTCCACG |
| L314A | for: TGACCGTGCTGGGATTCGATTGTATCACCACCGGCTACG<br>rev: CCCAGCACGGTCATGTACAGGAAGGCCGCGCCCATGC |
| L314S | for: TGACCGTGCTGGGATTCGATTGTATCACCACCGGCTACG<br>rev: CCCAGCACGGTCATGTACAGGAAGGCCGAGCCCATGC |
| L317A | for: TGACCGTGCTGGGATTCGATTGTATCACCACCGGCTACG<br>rev: CCCAGCACGGTCATGTACGCGAAGGCCAGGCCCATGC |
| L317S | for: TGACCGTGCTGGGATTCGATTGTATCACCACCGGCTACG<br>rev: CCCAGCACGGTCATGTACGAGAAGGCCAGGCCCATGC |
| Y318A | for: TGACCGTGCTGGGATTCGATTGTATCACCACCGGCTACG<br>rev: CCCAGCACGGTCATGGCCAGGAAGGCCAGGCCCATGC |
| Y318S | for: TGACCGTGCTGGGATTCGATTGTATCACCACCGGCTACG<br>rev: CCCAGCACGGTCATGGACAGGAAGGCCAGGCCCATGC |
| R466A | for: GAGTGATCGCCGCCGCAATCGGCCTGTGGTCCTTTGACCTGACAG<br>rev: GCGGCGATCACTCCAGCGAACAGGAGGCTCACCGAGATGATGG |
| L469A | for: GAGTGATCGCCGCCAGAATCGGCGCGTGGTCCTTTGACCTGACAG<br>rev: GCGGCGATCACTCCAGCGAACAGGAGGCTCACCGAGATGATGG |
| L469S | for: GAGTGATCGCCGCCAGAATCGGCTCGTGGTCCTTTGACCTGACAG<br>rev: GCGGCGATCACTCCAGCGAACAGGAGGCTCACCGAGATGATGG |
| W470A | for: GAGTGATCGCCGCCAGAATCGGCCTGGCGTCCTTTGACCTGACAG<br>rev: GCGGCGATCACTCCAGCGAACAGGAGGCTCACCGAGATGATGG |
| W470S | for: GAGTGATCGCCGCCAGAATCGGCCTGTCGTCCTTTGACCTGACAG<br>rev: GCGGCGATCACTCCAGCGAACAGGAGGCTCACCGAGATGATGG |
| D473A | for: GAGTGATCGCCGCCAGAATCGGCCTGTGGTCCTTTGCCCTGACAG<br>rev: GCGGCGATCACTCCAGCGAACAGGAGGCTCACCGAGATGATGG |
| Y501A | for: GAACAGCATGAACGCCCTGCTGGATCTGCTGCACTTCATCATGG<br>rev: GTTCATGCTGTTCTGCACGCCGTTGATGATTCCTCTCTCG |
| Y501S | for: GAACAGCATGAACTCCCTGCTGGATCTGCTGCACTTCATCATGG<br>rev: GTTCATGCTGTTCTGCACGCCGTTGATGATTCCTCTCTCG |
| F508A | for: GAACAGCATGAACTACCTGCTGGATCTGCTGCACGCCATCATGG<br>rev: GTTCATGCTGTTCTGCACGCCGTTGATGATTCCTCTCTCG |
| F508S | for: GAACAGCATGAACTACCTGCTGGATCTGCTGCACTCCATCATGG<br>rev: GTTCATGCTGTTCTGCACGCCGTTGATGATTCCTCTCTCG |

## Expression and purification of hsFPN

Suspension-adapted HEK293S GlnTI⁻ cells were grown in HyClone Trans FX-H media supplemented with 2% fetal bovine serum, 2 mM L-glutamine, 100 U/ml penicillin/streptomycin, 1 mM pyruvate, and 1.5 g/l kolliphor-P188 in a humified incubator at 37°C and 5% $CO_2$. Cells were checked for mycoplasma contamination. Plasmid DNA for transfection was amplified in *Escherichia coli* MC1061 and purified using a NucleoBond GigaPrep Kit. The day prior to transfection, 300 ml of suspension-adapted HEK293 GlnTI⁻ cells were seeded to 0.5 Mio/ml cell density. As transfection mixture, purified plasmid DNA was mixed with PEI MAX (PolySciences Inc) in a 1:2.5 (w/w) ratio and subsequently diluted with DMEM media (High-Glucose DMEM, Gibco, MERCK) to a final DNA concentration of 0.01 mg/ml.

The mixture was incubated at room temperature for 20 min prior to transfection, and each bioreactor vessel was supplemented with 50 ml transfection mixture and valproic acid to a final concentration of 4 mM. Expression was performed by incubation in a humidified incubator at 37°C, 5% $CO_2$ for 48 hr. Afterwards, the cells were harvested by centrifugation at 500 g, for 15 min at 4°C. The cell pellets were washed twice with PBS, flash-frozen in liquid nitrogen, and stored at –20°C until further use.

For purification, the cell pellet was thawed on ice and resuspended with ice-cold lysis buffer (20 mM Hepes pH 7.5, 200 mM NaCl, 2% [w/v] DDM, 10% w/v glycerol, 20 µg/ml DNase I, cOmplete protease inhibitor mix [Roche], 20 µl/ml Biotin blocking solution [IBA]). The suspension was sonified on ice in three cycles of 30 s using a VT70 titanium probe on a UW3200 sonicator. All further purification steps were performed at 4°C. Membrane proteins were extracted for 1 hr, while gently mixing. Cell debris and insoluble fractions were separated by centrifugation at 15,000 g for 25 min. The supernatant was loaded by gravity flow onto Strep-tactin Superflow resin (1.5–3 ml resin per liter of suspension culture) equilibrated with SEC buffer (20 mM Hepes pH 7.5, 200 mM NaCl, and 0.04% [w/v] DDM). The resin was washed with 20 column volumes (CV) SEC buffer, and bound proteins were eluted with five CV elution buffer (20 mM Hepes pH 7.5, 200 mM NaCl, 0.04% [w/v] DDM, and 5 mM Desthiobiotin). To cleave fusion tags, protein-containing fractions were supplemented with HRV-3C protease at a molar ratio of 5 (hsFPN) to 1 (3C protease) and incubated for 1 hr. The cleaved protein fraction was concentrated to approx. 500 µl using a 50 kDa cut-off centrifugal concentrator, filtered through a 0.22 µm filter, and finally injected into a SEC system connected to a Superdex S200 10/300 GL column, which was equilibrated with SEC buffer (20 mM Hepes pH 7.5, 200 mM NaCl, and 0.04% [w/v] DDM). For FP experiments, the DDM concentration in the SEC buffer was lowered to 0.02% (w/v) DDM. Peak fractions corresponding to monomeric FPN were pooled and concentrated to the desired concentration as described before.

## Expression and analysis of FPN mutants by fluorescence SEC

Adherent HEK293T cells were cultured in DMEM media supplemented with 10% fetal bovine serum and 100 U/ml penicillin/streptomycin in 100 mm TC dishes (Sarstedt AG) in an incubator at 37°C and 5% $CO_2$. Cells were checked for mycoplasma contamination and split the day prior to transfection to 30–40% confluence. The transfection mixes containing 10 µg plasmid and 40 µg polyethylenimine (PEI) in 1 ml of non-supplemented DMEM media were added to the adherent cells after 15 min of incubation at room temperature, together with 5 mM valproic acid. The cells were harvested after 2 days of expression, washed with 1 ml ice cold sterile PBS, and collected by centrifugation. Membrane proteins were extracted with 500 µl lysis buffer (20 mM Hepes pH 7.5, 200 mM NaCl, 2% [w/v] DDM, 10% w/v glycerol, 20 µg/ml DNase I, and protease inhibitor mix) for 1 hr under gentle mixing on a rotating wheel at 4°C. Insoluble fractions were separated by centrifugation at 15,000 g for 20 min at 4°C, and the supernatants were filtered through 0.22 µm sized pores. From the cleared flowthroughs, 50 µl were injected onto a Superdex S200 5/150 GL column equilibrated with gel-filtration buffer (20 mM Hepes pH 7.5, 200 mM NaCl, and 0.04% [w/v] DDM) connected to an HPLC system. EGFP fluorescence was excited at 488 nm, and the emission was recorded at 507 nm.

## Protein biotinylation

Purified protein at a concentration of 1 mg/ml was chemically biotinylated using EZ link NHS-PEG4-Biotin (Thermo Scientific, A39259) at a 10–30 times molar excess. The samples were incubated on ice for 1 hr, and the reaction was terminated by adding Tris-HCl pH 7.5 to a final concentration of 5 mM. Excess PEG-biotin was removed by SEC using a Superdex S200 10/300 GL column equilibrated with SEC buffer (20 mM Hepes pH 7.5, 200 mM NaCl, and 0.04% [w/v] DDM). The biotinylation level of hsFPN was estimated using mass spectroscopy and by incubation with an excess of Streptavidin followed by analysis of the resulting complexes by SDS-PAGE and SEC.

For SPR, purified FPN containing a C-terminal Avi-tag (30 µM) was enzymatically biotinylated in reaction buffer (20 mM HEPES pH 7.5, 200 mM NaCl, 0.04% DDM, 5 mM ATP, 10 mM magnesium acetate, 60 µM biotin, and 40 µg bifunctional ligase/repressor enzyme BirA). The sample was incubated overnight on ice and purified by SEC using a Superdex S200 10/300 GL column equilibrated with SEC buffer (20 mM Hepes pH 7.5, 200 mM NaCl, and 0.04% [w/v] DDM). The biotinylation efficiency was estimated as described above.

## Selection of synthetic nanobodies against hsFPN

Selection of synthetic nanobodies against hsFPN was performed as described previously (*Zimmermann et al., 2020*) with the detergent DDM added to a final concentration of 0.04% (w/v) to buffers used for membrane protein preparations. The selection was performed with mRNA libraries and vectors generously provided by Prof. Dr. Markus Seeger (Institute of Medical Microbiology, UZH). As a first step, one round of ribosome display was performed with the concave, loop, and convex mRNA libraries that each encode for around $10^{12}$ different binders with an hsFPN sample that was biotinylated to about 30%. After ribosome display, the eluted mRNA pool was cloned into a phagemid vector, and the resulting library was subsequently used in two rounds of phage display with a hsFPN sample that was 100% biotinylated at a molar ratio of PEG-biotin to hsFPN of two. During the second round of phage display, binders with high off rates were removed by incubation with non-biotinylated hsFPN at a concentration of 5 µM for 3 min. The concentrations of eluted phages were subsequently determined by quantitative PCR, and the specific enrichment was calculated by dividing the number of phages eluted in the hsFPN sample with the number of phages eluted in the negative control (biotinylated TM287/288; *Hutter et al., 2019*). Whereas, no enrichment could be detected for the concave and convex libraries (enrichment factors of 1.01 and 0.82, respectively), and the loop library resulted in an enrichment of 3.89. Thus, the DNA resulting from the loop library was further processed by cloning into the pSBinit vector and expression of the resulting sybodies in 96 well plates. Periplasmatic extractions were used for an ELISA assay in 384 well format, using either 50 µl of biotinylated hsFPN or biotinylated EcoDMT (*Ehrnstorfer et al., 2017*) as a control (at a concentration of 50 nM/well). In total, 42 ELISA hits were detected comprising 19 unique sybody sequences. All 19 binder candidates were further tested for expression and biochemical behavior. Subsequently, five sybodies (Sy1, Sy3, Sy8, Sy11, and Sy12) were further tested for binding to hsFPN by SEC and SPR.

## Expression and purification of sybodies

Plasmids encoding sybodies were transformed into *E. coli* MC1061, and single colonies were used for sybody expression in Terrific Broth media supplemented with 30 µg/ml chloramphenicol. For expression, media was inoculated with the preculture at a ratio of 1:100 (v/v). The culture was incubated at 37°C for 2 hr, and the temperature was subsequently lowered to 22°C. At an $OD_{600}$ 0.8–0.9, the culture was supplemented with L-arabinose at a final concentration of 0.02% (w/v) to induce sybody overexpression for 16–18 hr. The overnight culture was harvested at 4000 g for 20 min at 4°C. Pellets were flash frozen and stored at –20°C until further use.

For sybody purification, pellets from a 100 ml culture were resuspended in 10 ml periplasmatic extraction buffer (50 mM Tris-HCl pH 7.4, 20% [w/v] sucrose, 0.5 mM EDTA pH 8, 0.5 µg/ml lysozyme, and 20 µg/ml DNaseI) and incubated on ice for 30–60 min. The incubated mixture was diluted with 40 ml Tris-buffered saline (TBS) supplemented with 1 mM $MgCl_2$ and subsequently centrifuged at 4000 g for 20 min at 4°C to remove cell debris. The supernatant was supplemented with Ni-NTA resin (1 ml resin per 100 ml culture) and imidazole to a final concentration of 15 mM. The mixture was incubated for 1 hr under gentle agitation. After incubation, the resin was retained in a column and washed with 20 CV sybody wash buffer (TBS supplemented with 30 mM imidazole). The protein was eluted with five CV of sybody elution buffer (TBS supplemented with 300 mM imidazole). Protein containing fractions were pooled and concentrated to 500 µl using a 10 kDa cut-off concentrator. Concentrated protein was filtered through a 0.22 µm filter and injected into an Azura Knauer UVD 2.1 HPLC system connected to Sepax SRT 10 C-SEC100 column, which was equilibrated in sybody SEC buffer (20 mM Hepes pH 7.5 and 150 mM NaCl). Monomeric peak fractions were pooled and concentrated to 3–18 mg/ml as described before, flash frozen in liquid nitrogen, and stored at –80°C until further use.

## SPR binding assays

SPR experiments were performed using a BiaCore T200, with hsFPN immobilized on an SAD200M sensor chip for XanTec. For immobilization, hsFPN was expressed and purified with a C-terminal Avi-tag for enzymatic biotinylation. FPN at a concentration of 3 µg/ml was immobilized at a density of 675 RU. Flow cell 1 was left blank to serve as a reference cell for the measurements. Prior to measurements, the system was equilibrated for 2 hr with running buffer (20 mM HEPES pH 7.4, 200 mM NaCl, 0.04% DDM, and 0.1% BSA). All analytes were injected at 20°C at a flow rate of 30 µl/min

except for sybody 12, which was measured at a flow rate of 100 µl/min to exclude interference form mass transport effects. For the quantification, sybodies were injected at appropriate concentrations related to their binding affinities (sybody 1: 35.2, 70.5, 141, 282, 565, 1130, and 2260 nM; sybody 3: 3.4, 13.5, 54.5, 109, 217.5, 435, and 870 nM; sybody 8: 26, 52.5, 105, 210, 420, 840, and 1680 nM; sybody 11: 8.6, 34.5, 69, 137.5, 275, 550, and 1100 nM; sybody 12: 8, 32, 64, 128, 256, 512, and 1024 nM). Data was analyzed with the BIAevaluation software (GE Healthcare) and fitted to a single-site binding model. Data points influenced by machine noise were omitted from the analysis, and missing data are displayed as gaps. Very similar results were obtained for at least two independent protein preparations.

## SEC binding assays

For the binding assays, 30 µg hsFPN was incubated with vamifeport (Vifor Pharma) at a final concentration of 1 mM and incubated on ice for 5 min. In case of control experiments without vamifeport, this initial incubation step was left out. Incubated hsFPN was subsequently mixed with purified sybody in a 5× molar excess and incubated on ice for 30 min, filtered through 0.22 µm filters, and injected into HPLC system connected to a Superdex S200 5/150 GL column. Peak fractions of suspected monomeric hsFPN peaks were collected and concentrated to 40 µl (molecular weight cut-off 3 kDa). Evaluation of the binding assay was performed by SDS-PAGE analysis and comparison of peak height and retention volume of the suspected monomeric hsFPN peaks.

## Thermal stability assay using fluorescence-detection SEC

The assay was essentially performed as described (*Hattori et al., 2012*). Specifically, hsFPN aliquots of 30 µl at 0.5 µM, containing a 200-fold molar excess of vamifeport in indicated samples, were incubated at temperatures up to 75°C for 12 min. Aggregated protein was removed by centrifugal filtration (0.22 µm), and the resulting sample was subsequently subjected onto a Superose S6 column (GE Healthcare) equilibrated in SEC buffer. Proteins were detected using tryptophan fluorescence ($\lambda$ ex = 280 nm; $\lambda$ em = 315 nm) using a fluorescence detector (Agilent technologies 1200 series, G1321A). The peak heights of the monomeric hsFPN peaks were used to assess the stability by normalizing heights to the corresponding value from samples incubated at 4°C (100% stability). Melting temperatures ($T_m$) were determined by fitting the curves to a sigmoidal dose-response equation.

## Reconstitution of hsFPN into proteoliposomes

Purified hsFPN was reconstituted into detergent-destabilized liposomes according to the described protocol (*Geertsma et al., 2008*). Before reconstitution, soybean polar extract lipids (Avanti Polar lipids) were dried, washed with diethylether, and dried under nitrogen stream followed by exsiccation overnight. The dried lipids were then resuspended in liposome buffer (20 mM Hepes pH 7.5 and 150 mM KCl) in sonication cycles and flash frozen in liquid nitrogen three times before storing them at –80°C until further use.

For reconstitution, the thawed lipid stocks were extruded through a 400 nm filter (Liposo Fast-Basic, Avestin), and the extruded lipids were diluted to 4 mg/ml with liposome buffer. The diluted lipids were destabilized with Triton-X100 while monitoring light scattering at 540 nm. For reconstitutions, protein to lipid ratio of either 1:70 (w/w) or 1:50 (w/w) was used. After addition of the protein to the lipids, the mixture was incubated for 15 min at room temperature while gently mixing. Subsequently, detergent was removed by excessive addition of BioBeads SM-2 (BioRad) over the course of 3 days.

After removal of detergent, the proteoliposomes were harvested by centrifugation at 236,400 g for 40 min at 16°C. After centrifugation, the proteoliposomes were resuspended in liposome buffer to a final concentration of 20 mg/ml, flash frozen in liquid nitrogen, and stored at –80°C until further use.

## Fluorescence-based substrate transport assays

To measure the $Co^{2+}$ transport into liposomes mediated by hsFPN, 1 mg of thawed proteoliposomes were mixed with 400 µl buffer IN (20 mM Hepes pH 7.5, 100 mM KCl, and 250 µM calcein). After five freeze-thaw cycles in liquid nitrogen, the liposomes were extruded through a 400 nm filter, harvested by centrifugation at 170,000 g for 25 min at 22°C, and subsequently washed twice with buffer WASH (20 mM Hepes pH 7.5 and 100 mM KCl). The washed liposomes were resuspended with buffer WASH to a final lipid concentration of 25 mg/ml. The assay was started by diluting the liposome stock into

buffer OUT (20 mM Hepes pH 7.5 and 100 mM NaCl) to a final concentration of 0.25 mg/ml and distributing aliquots of 100 μl in a black 96-well plate (Thermo Fischer Scientific). The calcein fluorescence was measured in 4 s intervals on Tecan Infinite M1000 fluorimeter with the excitation wavelength $\lambda$ ex = 492 nm, and the emission wavelength $\lambda$ em = 518 nm until a stable signal was obtained. The addition of valinomycin (Invitrogen) to a final concentration of 100 nM established a negative membrane potential of –118 mV as consequence of 100-fold outwardly directed $K^+$ gradient. During transport experiments, the established membrane potential did not affect transport rates as expected for an electroneutral process. To start transport, $CoCl_2$ was added in different concentrations to the liposome suspension. As final step, liposomes were supplemented with calcimycin (Invitrogen) to a final concentration of 100 nM to equilibrate $Co^{2+}$ concentrations. Transport data were analyzed in GraphPad Prism 8.4.3. (GraphPad Software, LLC).

## FP assays

The FP assays were performed in a similar way as described (*Manolova et al., 2019*). In these assays, a derivative of the peptide hormone hepcidin was used that was labeled with TMR at position 21, where the Met was replaced with a Lys. To determine direct binding of TMR-hepcidin to hsFPN, purified WT or mutant proteins serially diluted in FP assay buffer containing 20 mM Hepes pH 7.4, 200 mM NaCl, 0.02% DDM, and 0.1 mg/ml BSA were plated into a 384-well black low-volume round bottom plate (Corning) at 16 μl per well. The final FPN concentrations ranged from 13 nM to 6 μM. TMR-hepcidin was added at a volume of 8 μl to reach a final concentration of 10 nM. To determine unspecific binding, 30 μM vamifeport was added to the reaction mixture. For displacement assays, a mixture of purified WT hsFPN or its point mutants and 15 nM TMR-hepcidin in FP assay buffer was plated into a 384-well black low-volume round bottom plate (Corning) at 16 μl per well. Competitors (vamifeport, hepcidin-25 [Bachem] or hepcidin-20 [Bachem]) were added at a volume of 8 μl per well from serial dilutions to reach a final TMR-hepcidin concentration of 10 nM. Final hsFPN concentrations varied depending on the measured affinity for TMR-hepcidin to the corresponding protein constructs. Specifically, for the results displayed in *Figure 5*, the final hsFPN concentrations were 400 nM for WT, L469A, L469S, and W470S and 800 nM for R466A. For both types of experiments, direct binding and displacement assays, the plates were incubated at room temperature for 90 min, and subsequently, the parallel ($F_{para}$) and perpendicular ($P_{perp}$) fluorescence were measured in a Synergy H1 fluorescence reader (BioTek). The following formula was used to calculate the FP in mP:

$$FP = \frac{Fpara - Fperp}{Fpara + Fperp} \cdot 1000$$

To calculate $K_D$ values of hsFPN to TMR-hepcidin, FP data was fitted to the following equation:

$$Y = B_{max} \frac{x}{x + K_D} + NS \cdot x + background$$

With Y being the FP value, x the variable hsFPN concentration, $K_D$ the dissociation constant, NS the slope of the non-specific binding signal, and $B_{max}$ the maximal specific binding signal. $B_{max}$ and the background signal were constrained to the same values for WT and mutants. The fitting yielded $K_D$ values of 100±4 nM for WT, 57±2 nM for L469A, 94±3 nM for L469S, 227±8 nM for W470S, 501±22 nM for R466A, 1044±60 nM for V68S, and about 4.5 μM and 6 μM for Y501S and D504A, respectively.

To obtain $IC_{50}$ values from displacement data, FP data was fitted to a four parameter Hill equation.

$$Y = Bottom + \frac{Top - Bottom}{1 + 10^{(logIC_{50} - x) \cdot n}}$$

With Y being the FP value, x the variable competitor concentration, and n the Hill coefficient. Bottom and Top FP values were constrained to the same values for hepcidin-25 and vamifeport (225 mP and 307 mP, respectively).

To convert obtained $IC_{50}$ values to $K_D$s, the displacement data were fitted to a competition binding model:

$$Y = \frac{L_{dis}}{L_{dis} + K_{D(dis)} \cdot \left(1 + \frac{x}{K_{D(x)}}\right)}$$

With Y being the fractional saturation of TMR-hepcidin binding to hsFPN, which was calculated using the affinity of TMR-hepcidin to hsFPN measured before. $L_{dis}$ corresponds to the concentration of the displaced ligand (400 nM FPN for WT, L469A, L469S, and W470S and 800 nM for the R466A mutant), $K_{D(dis)}$ is the dissociation constant for the FPN/TMR-hepcidin interaction measured before, x is the variable competitor concentration, and $K_{D(x)}$ is the fitted dissociated constant for the FPN/competitor interaction. For $K_{D(hep-25)}$ values of 131±12 nM for WT, 108±17 nM for L469A, 180±17 nM for L469S, 222±19 nM for W470S, and 713±117 nM for R466A were obtained. For $K_{D(vamifeport)}$ values of 24±3 nM for WT, 37±4 nM for L469A, 60±7 nM for L469S, 82±11 nM for W470S, and 83±16 nM for R466A were obtained.

All fits were performed in GraphPad Prism 8.4.3. (GraphPad Software, LLC). For all FP experiments, very similar results were obtained for at least two different protein preparations and at different concentrations of hsFPN.

## Preparation of the hsFPN complexes for cryo-EM and data collection

Samples of freshly purified hsFPN at a concentration of 3–4 mg/ml were incubated with vamifeport at a final concentration of 1 mM on ice for 5 min prior to further supplementing it with a 1.5× molar excess of sybody 3 or 12. For sample without vamifeport, hsFPN was only supplemented with sybody 3. For all conditions, the hsFPN concentration in the final sample was between 2.5 and 3 mg/ml, a final detergent concentration of 0.04% (w/v) DDM was maintained, and all complexes were incubated for at least 30 min on ice prior to grid freezing.

After incubation on ice, 2.5 µl of complex mixture was applied on glow-discharged holey carbon grids (Quantifoil R1.2/1.3, Au 200 mesh), and excess of liquid was removed by blotting for 2–4 s in controlled environment (4 °C, 100% humidity) using a Vitrobot Mark IV. Samples were plunge-frozen in a propane-ethane freezing mixture and stored in liquid nitrogen until data acquisition.

All datasets were collected on a 300 kV Titan Krios G3i (ThermoFischer Scientific) with a 100 µm objective aperture and using a post-column BioQuantum energy filter (Gatan) with a 20 eV slit and a K3 direct electron detector (Gatan) operating in a super-resolution mode. Micrographs were recorded in an automated manner using EPU2.9 with a defocus range from –1 to –2.4 µm at a magnification 130,000× corresponding to a pixel size of 0.651 Å per pixel (0.3255 Å in super resolution mode) and an exposure of 1.01 s (36 frames) and a dose of approximately 1.8 e⁻/Å²/frame. The total electron dose on the specimen level for all datasets was between 61 e⁻/Å² and 71 e⁻/Å².

## Cryo-EM image processing

All datasets were pre-processed in the same manner (*Figure 2—figure supplements 1–3*) using cryoSPARC v.3.0.1, v3.2.0, and v4.0.3 (*Punjani et al., 2017*). Micrographs were subjected to patch motion correction with a fourier crop factor of 2 (pixel size of 0.651 Å /pix) followed by patch CTF estimation. Based on CTF estimations, low-quality micrographs showing a significant drift, ice contamination, or poor CTF estimates were discarded resulting in datasets of 4384 images of hsFPN/Sy3 complex (dataset 1), 8752 images of hsFPN/Sy3 complex with vamifeport (dataset 2), and 13,730 images of hsFPN/Sy12 complex with vamifeport (dataset 3), which were subjected to further data processing. Particles were initially picked using a blob picker with a minimum particle diameter of 120 Å and a minimum inter-particle distance of 60 Å. Selected particles were extracted with a box size of 360 pixels (down-sampled to 180 pixels at a size of 1.302 Å/pixel) and subjected to 2D classification. 2D class averages showing protein features were subsequently used as templates for more accurate template-based particle picking as well as inputs for generating two 3D ab initio models. Subsequently, the promising 2D classes were subjected to several rounds of heterogenous refinements using one of the ab initio models as a 'reference' and an obviously bad model as a decoy model. After several rounds of heterogenous refinement, the selected particles and models were subjected to non-uniform refinement (input model lowpass-filtered to 15 Å). The hsFPN/Sy3 complex with vamifeport was additionally subjected to CTF-refinement followed by a second round of non-uniform refinement. The quality of the map was validated using 3DFSC (*Tan et al., 2017*) for FSC validation and local resolution estimation. For Bayesian polishing, the particles of the respective best maps of all three datasets were exported from cryoSPARC and converted into a RELION 4.0 readable star file using the cspar2star script of pyem (*Asarnow et al., 2019*). The raw micrographs were down-sampled twice, and the beam-induced motion in the movies was corrected in all frames using RELION's implementation of

MotionCor2 (*Zheng et al., 2017*). The contrast transfer function parameters were estimated using CTFFIND4 (*Rohou and Grigorieff, 2015*). The particles were imported to RELION and re-extracted from the two-fold binned motion corrected micrographs (pixel size of 0.651 Å /pix) using a box size of 360 pixels. The extracted particles were imported back to cryoSPARC and associated with the RELION 4.0 motion corrected micrographs with estimated CTF parameters and were subjected to a non-uniform refinement. The resulting maps and masks were used for a round of 3D refinement and postprocessing in RELION (*Kimanius et al., 2016*; *Scheres, 2012*). The particles were polished (*Zivanov et al., 2019*) using an expected 0.2 Å/dose movement, 5000 Å spatial correlation length, and an average particle acceleration of 2 Å/dose. The polished particles were imported back to cryo-SPARC and associated with the RELION motion corrected micrographs and subjected to a final non-uniform refinement. This procedure resulted in an improvement of the resolution of the hsFPN/Sy3/vamifeport dataset from 3.37 to 3.24 Å (*Figure 2—figure supplement 2*) but not of the hsFPN/Sy12/vamifeport and the hsFPN/Sy3 datasets.

## Model building and refinement

The models of the hsFPN/Sy3 complex with and without vamifeport and the hsFPN/Sy12 complex with vamifeport were built in Coot (*Emsley and Cowtan, 2004*), using the published human FPN structures as references (*Billesbølle et al., 2020*; *Pan et al., 2020*). Vamifeport was generated using the ligand tool implemented in Coot, and constraints for the refinement were generated using the CCP4 program PRODRG (*Schüttelkopf and van Aalten, 2004*). The model was improved iteratively by cycles of real-space refinement in PHENIX (*Adams et al., 2002*) with secondary structure constraints applied followed by manual corrections in Coot. Validation of the model was performed in PHENIX. Surfaces were calculated with MSMS (*Sanner et al., 1996*). Figures containing molecular structures and densities were prepared with DINO (http://www.dino3d.org), Chimera (*Pettersen et al., 2004*), and ChimeraX (*Pettersen et al., 2021*).

## Docking

The interaction energy between hsFPN and vamifeport in the two possible binding orientations (original and alternative, *Figure 4—figure supplement 3*) was estimated with a docking protocol for predicting binding affinities based on the ABSINTH implicit solvation model (*Marchand et al., 2020*; *Vitalis and Pappu, 2009*) and implemented in the CAMPARI v4 software package (https://campari.sourceforge.net/V4/). Each run consisted of 25,000 steps of hybrid Monte Carlo (MC) and Molecular Dynamics (MD; 4000 steps of MC, followed by alternating cycles of 200 steps of MD and 50 steps of MC) in hybrid rigid body and torsional space (*Vitalis and Pappu, 2014*). The simulations were carried out in the NVT ensemble at 250 K, with velocity-rescaling thermostat (0.1 ps coupling time; *Bussi et al., 2007*).

In order to preserve the pose of vamifeport, the non hydrogen atoms of the ligand were harmonically restrained to their initial positions (with a force constant of 0.1 kcal/mol/$Å^2$), and its internal degrees of freedom were frozen, only allowing rigid body rotations and translations. For hsFPN, atoms within a maximum distance of 15 Å from vamifeport were allowed to move to permit local adaptations to both investigated binding orientations of the inhibitor. As exception, the coordinates of Asp 259 were fixed to prevent its reorientation to optimize its interactions with the protonated amine of the linker in the alternative orientation of vamifeport. This reorientation does not match the experimental cryo-EM density and thus presents additional evidence that the alternative pose is less compatible with the observed protein conformation. In our case, this meant that 16 side chain dihedral angle degrees of freedom in 8 distinct binding site residues were allowed to move, where those involving only hydrogen atoms (like R-C-O-H) are also included in the count. The system was parameterized with ABSINTH 4.2 together with CHARMM 36 force field (*Best et al., 2012*), and the atomic charges of vamifeport were generated with CGenFF v2.5.1 (*Vanommeslaeghe and MacKerell, 2012a*; *Vanommeslaeghe et al., 2012b*). For each of the two alternative vamifeport orientations, 20 simulations were run, and the pose with the minimum single-point energy was selected (*Figure 4—figure supplement 4*). In the ABSINTH model, the total internal energy is the sum of bonded terms, a van der Waals term, and a solvation term encompassing both the direct mean field interaction (DMFI) between solute and solvent and screened polar interactions. The DMFI is based on experimental-free energies of solvation and thus accounts for entropic, polar, and non-polar terms. We report both the

total energy (after subtraction of the energetic penalty due to restraints) and the solvation and van der Waals (vdW) terms separately. The mean energy penalty for restraining the ligand was 1.12 kcal/mol for poses in the original orientation and 3.62 kcal/mol for poses in the alternative orientation of vamifeport (*Figure 4—figure supplement 4A, B*), which further demonstrates that the original binding position fits the protein conformation better, as unrestrained runs would deviate much more from the density in the alternative case. Deviating from the published scoring protocol (*Marchand et al., 2020*), we were only interested in the difference in binding affinity between the two orientations of vamifeport. Hence, we directly compared the total energies of the complex. In the docking runs, the original conformation described in this study consistently had a lower energy by about 12–20 kcal/mol compared to the best poses for the alternative conformation, where the difference in energy is dominated by solvation and polar terms (*Figure 4—figure supplement 4C, D*).

## Acknowledgements

This research was supported by the Swiss National Science Foundation (SNF) through the National Center of Competence in Research TransCure. We thank Dr. Marta Sawicka for input in cryo-EM and help during initial sample characterization and Dr. Andreas Vitalis for help with docking calculations. The cryo-electron microscope and K3-camera were acquired with support of the Baugarten and Schwyzer-Winiker foundations and a Requip grant of the Swiss National Science Foundation. We thank Prof. Markus Seeger for access to the sybody libraries, Dr. Jens Sobek from the Functional Genomics Center Zurich for input and assistance for SPR measurements, the center for Microscopy and Image Analysis (ZMB) of the University of Zurich for their support and access to the electron microscope, and Rachelle Gaudet for providing the coordinates of the occluded conformation of DraNRAMP prior to their release. All members of the Dutzler lab are acknowledged for help in various stages of the project.

## Additional information

### Competing interests

Patrick Altermatt: P.A. is employee of CSL Vifor and is inventor in patents related to the publication (WO2021013771A1). Vania Manolova: V.M. is employee of CSL Vifor and is inventor in patents related to the publication (WO2017068089A9, WO2017068090, WO2021013771A1, WO2021013772A1, WO2021078889A1, WO2022157185A1). Hanna Sundstrom: H.S. is employee of CSL Vifor. Franz Dürrenberger: F.D. is employee of CSL Vifor and is inventor in patents related to the publication (WO2017068089A9, WO2017068090, WO2021013771A1, WO2021013772A1, WO2021078889A1, WO2022157185A1). The other authors declare that no competing interests exist.

### Funding

| Funder | Grant reference number | Author |
| --- | --- | --- |
| Schweizerischer Nationalfonds zur Förderung der Wissenschaftlichen Forschung | NCCR TransCure | Raimund Dutzler |

The funders had no role in study design, data collection and interpretation, or the decision to submit the work for publication.

### Author contributions

Elena Farah Lehmann, Formal analysis, Validation, Investigation, Methodology, Writing – review and editing, cloned; expressed and purified proteins; prepared samples for cryo-em; processed cryo-em data; built models and performed proteoliposome-based transport assays; Márton Liziczai, Formal analysis, Validation, Investigation, Methodology, Writing – review and editing, Cloned, expressed and purified proteins, prepared samples for cryo-EM, processed cryo-EM data, built models and performed SPR experiments; Katarzyna Drożdżyk, Data curation, Formal analysis, Validation,

Investigation, Methodology, Writing – review and editing, Collected cryo-EM data; Patrick Altermatt, Conceptualization, Formal analysis, Investigation, Methodology, Writing – review and editing, Performed FP experiments; Cassiano Langini, Conceptualization, Software, Supervision, Investigation, Methodology, Project administration, Writing – review and editing, Performed docking calculations; Vania Manolova, Conceptualization, Formal analysis, Supervision, Investigation, Methodology, Project administration, Writing – review and editing; Hanna Sundstrom, Conceptualization, Formal analysis, Supervision, Investigation, Methodology, Project administration, Writing – review and editing, Performed FP experiments; Franz Dürrenberger, Conceptualization, Supervision, Funding acquisition, Validation, Visualization, Writing – original draft, Project administration, Writing – review and editing; Raimund Dutzler, Conceptualization, Data curation, Formal analysis, Supervision, Funding acquisition, Validation, Investigation, Visualization, Methodology, Writing – original draft, Project administration, Writing – review and editing; Cristina Manatschal, Conceptualization, Data curation, Formal analysis, Supervision, Validation, Investigation, Visualization, Methodology, Writing – original draft, Project administration, Writing – review and editing, cloned; expressed and purified proteins; prepared samples for cryo-em; processed cryo-em data; built models and performed proteoliposome-based transport assays

## Author ORCIDs
Elena Farah Lehmann http://orcid.org/0000-0002-6112-933X
Márton Liziczai http://orcid.org/0000-0002-6673-9209
Katarzyna Drożdżyk http://orcid.org/0000-0001-6288-4735
Franz Dürrenberger http://orcid.org/0000-0002-1443-1181
Raimund Dutzler http://orcid.org/0000-0002-2193-6129
Cristina Manatschal http://orcid.org/0000-0002-4907-7303

## Decision letter and Author response
Decision letter https://doi.org/10.7554/eLife.83053.sa1
Author response https://doi.org/10.7554/eLife.83053.sa2

## Additional files

### Supplementary files
• MDAR checklist

### Data availability
The cryo-EM density maps of the hsFPN/Sy3 complexes in absence and presence of vamifeport and of hsFPN/Sy12 in presence of vamifeport have been deposited in the Electron Microscopy Data Bank under ID codes EMD-16353 and EMD-16345 and EMD-16354, respectively. The coordinates for the atomic models of hsFPN/Sy3 in absence of vamifeport refined against the 4.09 Å cryo-EM density, hsFPN/Sy3 in presence of vamifeport refined against the 3.37 Å cryo-EM density and hsFPN/Sy12 in presence of vamifeport refined against the 3.89 Å cryo-EM density have been deposited in the Protein Data Bank under ID codes 8C02 and 8BZY and 8C03, respectively. Source data files have been provided for Figure 1, Figure1-figure supplement 1, Figure1-figure supplement 2, Figure 1-figure supplement 3, Figure 4-figure supplement 4, Figure 5, Figure 5-figure supplement 1, Figure 5-figure supplement 2.

The following datasets were generated:

| Author(s) | Year | Dataset title | Dataset URL | Database and Identifier |
|---|---|---|---|---|
| Lehmann EF, Liziczai M, Drozdzyk K, Dutzler R, Manatschal C | 2023 | Structure of SLC40/ ferroportin in complex with synthetic nanobody Sy3 in occluded conformation. | https://www.ebi.ac. uk/emdb/EMD-16353 | Electron Microscopy Data Bank, EMD-16353 |

*Continued on next page*

*Continued*

| Author(s) | Year | Dataset title | Dataset URL | Database and Identifier |
|---|---|---|---|---|
| Lehmann EF, Liziczai M, Drozdzyk K, Dutzler R, Manatschal C | 2023 | Structure of SLC40/ferroportin in complex with vamifeport and synthetic nanobody Sy3 in occluded conformation | https://www.ebi.ac.uk/emdb/EMD-16345 | Electron Microscopy Data Bank, EMD-16345 |
| Lehmann EF, Liziczai M, Drozdzyk K, Dutzler R, Manatschal C | 2023 | Structure of SLC40/ferroportin in complex with vamifeport and synthetic nanobody Sy12 in outward-facing conformation. | https://www.ebi.ac.uk/emdb/EMD-16354 | Electron Microscopy Data Bank, EMD-16354 |
| Lehmann EF, Liziczai M, Drozdzyk K, Dutzler R, Manatschal C | 2023 | Structure of SLC40/ferroportin in complex with synthetic nanobody Sy3 in occluded conformation. | https://www.rcsb.org/structure/8C02 | RCSB Protein Data Bank, 8C02 |
| Lehmann EF, Liziczai M, Drozdzyk K, Dutzler R, Manatschal C | 2023 | Structure of SLC40/ferroportin in complex with vamifeport and synthetic nanobody Sy3 in occluded conformation. | https://www.rcsb.org/structure/8BZY | RCSB Protein Data Bank, 8BZY |
| Lehmann EF, Liziczai M, Drozdzyk K, Dutzlar R, Manatschal C | 2023 | Structure of SLC40/ferroportin in complex with vamifeport and synthetic nanobody Sy12 in outward-facing conformation. | https://www.rcsb.org/structure/8C03 | RCSB Protein Data Bank, 8C03 |

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
