## [Editor Report]

This important study reports cryo-EM structures of human ferroportin (FPN), a protein essential for iron transport in humans, and will be of interest to researchers studying membrane transport mechanisms as well as to those interested in drug design. Structures detail interactions between FPN and the small-molecule inhibitor vamifeport, which is currently in clinical trials for sickle cell disease. In addition, they identify a new (occluded) protein conformation, stabilized by a small-protein binder, that may be relevant to the transport mechanism. Evidence for the mechanism of inhibition by vamifeport is convincing.

---

## [Decision Letter]

**Decision letter after peer review:**

Thank you for submitting your article "Structures of ferroportin in complex with its specific inhibitor vamifeport" for consideration by *eLife*. Your article has been reviewed by 3 peer reviewers, and the evaluation has been overseen by a Reviewing Editor and Richard Aldrich as the Senior Editor. The following individual involved in the review of your submission has agreed to reveal their identity: Rachelle Gaudet (Reviewer #3).

The reviewers have discussed their reviews with one another, and the Reviewing Editor has drafted this letter to help you prepare a revised submission. Overall, the reviewers appreciate the quality of your data but have concerns about presentation and interpretation. The recommendations below are mostly focused on improving the clarity of presentation and appropriateness of conclusions.

Essential revisions:

1) The formation of the occluded state may just be a consequence of Sy3 binding. Directly testing whether this state exists during the transport cycle when the protein is unconstrained could be extremely difficult but is essential for concluding whether the state is mechanistically relevant. At a minimum, the authors should discuss this point.

2) The authors state that several of the mutations negatively impact the expression levels of ferroportin. They should show some data to support this statement (at a minimum, a Western blot).

3) A major concern is the disconnect between the Results section of the paper and the Discussion. The current version appears more focused on fleshing out the transport mechanism of FPN rather than the mechanism of vamifeport inhibition. One way to address this issue would be to rewrite the paper to focus on either the vamfeport mechanism or the transport mechanism. Given the concern raised in point 1, if the transport mechanism remains prominent, it would be essential to add more experimental results to support this aspect of the Discussion section. Regardless, we would like to see more focus on the mechanism of inhibition in the Discussion section and more on the implications of the mutational data presented in Figure 5.

4) Also concerning the discussion: There is a lot of detail concerning the S1 and S2 sites, which are not well introduced in the Results section, and what relevance do these sites, or this Discussion, have with the mechanism of vamifeport inhibition? Similarly. Line 354 discusses the metal ion binding site, which seems out of place, given this appears to have nothing to do with vamifeport binding. This section appears disjointed.

5) It would enhance the discussion of the vamifeport binding site to provide information about whether the residues are conserved in ferroportin homologs. For example, vamifeport is effective in mouse models: are the binding residues conserved between human and mouse (especially the ones that may make specific polar interactions)? Also, is there a correlation between the level of conservation and the residues that reduce protein expression?

6) The figures comparing the two possible inhibitor orientations are a bit difficult to interpret to independently assess the orientation chosen by the authors. Could the authors also provide some more quantitative assessment, for example, by comparing the docking energy of different alternative poses?

7) It would be useful to list the measured Kd values of the various ferroportin mutants for hepcidin and Vamifeport in a table, for ease of reference and comparison.

8) The authors may want to update the comparison with the occluded structure of DrNramp with this recent BioRxiv preprint: doi: https://doi.org/10.1101/2022.09.08.507188.

9) Are the interactions between vamifeport and FPN similar to that observed for Hepcidin? Do the interactions explain the increased affinity observed for vamifeport over Hepcidin?

10) Figure 4 supplement 3B should include the interactions, possibly as a schematic.

11) Lines 268-271 refer to the analysis of the metal binding site and speculation that the conformation stabilized by the Sy3 binder precludes the simultaneous binding of both vamifeport and metal ion. However, physiologically, can the authors demonstrate whether vamifeport prevents Fe binding?

12) Lines 315-316 refer to the two major advances reported in this study: the structures of FPN in the occluded state and the subsequent complex and functional characterization of vamifeport binding. However, it is unclear to what extent the structures are related to physiological relevant states, given that the sybodies appear to stabilize the metal ion binding site in a conformation that cannot now bind metal ions. This section needs more explanation.

13) Lines 329 – 331, we think the first study to reference the role of helix 7 in the MFS as the 'gating helix' was the 2003 GlpT paper (10.1126/science.1087619), followed by the FucP (10.1038/nature09406) and POT (doi.org/10.1038/emboj.2010.309) studies.

14) Lines 332-334 – these lines describe what appears to be a significant mechanistic feature, that vamifeport binding causes TM7 to rigidify, which might also be a component of the inhibition mechanism. Why is this not clearly described in the Results section?

15) Lines 339 – the authors appear to focus a great deal of comparative analysis between FPN and the GLUT sugar transporters. Why? It would seem to me a more similar system would be by the multidrug exporters or peptide transporters, which also contain multiple binding sites. The link between FPN and GLUTs is not well described to make this comparison obvious.

---

## [Author Response]

Essential revisions:1) The formation of the occluded state may just be a consequence of Sy3 binding. Directly testing whether this state exists during the transport cycle when the protein is unconstrained could be extremely difficult but is essential for concluding whether the state is mechanistically relevant. At a minimum, the authors should discuss this point.2) The authors state that several of the mutations negatively impact the expression levels of ferroportin. They should show some data to support this statement (at a minimum, a Western blot).

We have discussed the ambiguity concerning the functional correspondence of the observed occluded state and removed claims that classified it as transport intermediate throughout the manuscript. However, we also mentioned that this conformation is likely populated in solution since sybodies frequently target preexisting conformations and since selections were performed in absence of vamifeport.

Line 343-353:

Since this conformation with occluded substrate binding pocket is stabilized by the binding of Sy3 from the extracellular side to a cleft between both sub-domains involving α-helices 1, 5, 7 and 8 (Figure 2D), it is currently not known to which degree it represents an intermediate in the transport cycle. Notwithstanding the ambiguity concerning its functional relevance, several lines of evidence suggest that this conformation would be assumed in solution and at least be close to conformations on the transition towards the inward-facing state. In the sybody selection process, binders frequently target pre-existing conformations or ones that are energetically close. Consequently, we find it unlikely that this conformation would be solely induced by sybody binding, particularly since it is also targeted by vamifeport, which was not present during the selection process.

We have presented the relative expression of mutants as a new Figure (Figure 5—figure supplement 1).

3) A major concern is the disconnect between the Results section of the paper and the Discussion. The current version appears more focused on fleshing out the transport mechanism of FPN rather than the mechanism of vamifeport inhibition. One way to address this issue would be to rewrite the paper to focus on either the vamfeport mechanism or the transport mechanism. Given the concern raised in point 1, if the transport mechanism remains prominent, it would be essential to add more experimental results to support this aspect of the Discussion section. Regardless, we would like to see more focus on the mechanism of inhibition in the Discussion section and more on the implications of the mutational data presented in Figure 5.

We have restructured the discussion to put the main focus on the interaction with vamifeport. We have also emphasized that the mechanism of how vamifeport inhibits iron transport is still poorly understood and that this could either be accomplished by the lock of FPN in the observed occluded conformation to directly block transport or, alternatively, by the promotion of its internalization. With respect to vamifeport binding, we have better documented the overlap with the hepcidin site. However, this comparison is limited by the different protein conformations stabilized by both molecules. While the mutation of residues in the part of the binding site that is not overlapping with the hepcidin site confirm our described binding mode, there is little mechanistic insight beyond that.

While we have strongly reduced the discussion related to the transport mechanism, we regard the unusual features concerning the spacious occluded pocket and the two disconnected metal ionbinding sites as important characteristic features of FPN that deserve to be addressed. Although we do not claim that the described occluded conformation would be a true intermediate on the transport cycle, the comparison of inward and outward-facing conformations of a bacterial FPN homologue independently suggest the formation of a spacious aqueous pocket during transport. With respect to vamifeport binding, this is noteworthy since the compound only occupies a third of the binding pocket, which implies that there would be still ample of space for potential modifications.

4) Also concerning the discussion: There is a lot of detail concerning the S1 and S2 sites, which are not well introduced in the Results section, and what relevance do these sites, or this Discussion, have with the mechanism of vamifeport inhibition? Similarly. Line 354 discusses the metal ion binding site, which seems out of place, given this appears to have nothing to do with vamifeport binding. This section appears disjointed.

We have removed the sentence in line 345, introduced the S1 and S2 binding sites in the introduction in more detail and described conformational differences of these sites in our current structures in the results. Although the detailed transport mechanism of FPN has remained to some degree elusive, the way how the protein captures its substrate strongly differs from other well characterized transition metal ion transporters. We believe that these features deserve to be mentioned, in light of the spacious substrate binding pocket found in the occluded conformation. The importance of the metal ion binding sites for vamifeport interactions is evident since the compound occupies one of these sites (S2) but does not interfere with the other (S1). This property is described in the results and illustrated in Figure 4—figure supplement 4.

Line 85-91:

Together, these studies suggest that, distinct from other MFS-transporters, FPN likely contains two conserved substrate binding sites. The first site resides within the N-domain (S1) and comprises an aspartate and a histidine on α1 (*i. e.* Asp39 and His 43) in both hsFPN and tsFPN. A metal ion was also found to bind to a close by location between α1 and α6 in the prokaryotic bbFPN. A second site that is confined to the mammalian transporters is contained within the C-domain (S2) and briges a conserved cysteine (Cys 326) on α7 with a histidine on α11 (*i.e.* His 507 in hsFPN and His 508 in tsFPN).

Line 263-272:

Besides overlapping with the hepcidin binding site, vamifeport is located in close proximity to the S2 metal binding site with His 507 on α11 and Cys 326 on the α7a-α7b loop acting as coordinating residues (Figure 4F, Figure 4—figure supplement 5A). In all present mammalian FPN structures, the position of His 507 displays only small variations, whereas the position of Cys 326 on the flexible loop shows large differences (Figure 4—figure supplement 5A, B). In presence of bound metal ions, His 507 and Cys 326 are in about 3.5 Å distance to each other, while in the same outward-facing conformation in absence of substrates, the distance to His 507 increases to 6 Å. In case of both hsFPN/Sy3 structures Cys 326 moves even further towards the N-terminal domain and enlarges the distance to His 507 to 7.5 Å, which likely prohibits the concomitant binding of vamifeport and a divalent metal ion at the S2 site (Figure 4—figure supplement 5A, B).

5) It would enhance the discussion of the vamifeport binding site to provide information about whether the residues are conserved in ferroportin homologs. For example, vamifeport is effective in mouse models: are the binding residues conserved between human and mouse (especially the ones that may make specific polar interactions)? Also, is there a correlation between the level of conservation and the residues that reduce protein expression?

The residues in contact with vamifeport are strongly conserved within the SLC40 family and identical between mammalian members. We have shown an alignment of the sequences surrounding the binding region in Figure 2—figure supplement 5B and added the following sentence.

Line 372-374:

These residues are highly conserved throughout FPN orthologs and they are identical in all mammalian family members (Figure 2—figure supplement 5B), explaining the inhibitory action of vamifeport in mice.

6) The figures comparing the two possible inhibitor orientations are a bit difficult to interpret to independently assess the orientation chosen by the authors. Could the authors also provide some more quantitative assessment, for example, by comparing the docking energy of different alternative poses?

In our revision, we have estimated the interaction of the inhibitor with FPN in both orientations by computational docking and find a more favorable interaction energy in the proposed ‘original orientation’. We have mentioned this in the text, added a chapter in the method section and provided a novel Figure supplement (Figure 4—figure supplement 4).

Line 270-272:

The preferred interactions in the initially described orientation of vamifeport are also reflected in the 12-20 kcal/mol lower docking energy of the compound, which is dominated by the more favorable polar interactions established with the protein (Figure 4—figure supplement 4).

The docking protocol is described in detail in the methods Line 770-813

7) It would be useful to list the measured Kd values of the various ferroportin mutants for hepcidin and Vamifeport in a table, for ease of reference and comparison.

We have listed the values in the new Table 2.

8) The authors may want to update the comparison with the occluded structure of DrNramp with this recent BioRxiv preprint: doi: https://doi.org/10.1101/2022.09.08.507188.

We have referred to this preprint and provide an updated panel in Figure 6—figure supplement 1F.

9) Are the interactions between vamifeport and FPN similar to that observed for Hepcidin? Do the interactions explain the increased affinity observed for vamifeport over Hepcidin?

We have extended the comparison of the interactions of FPN with either hepcidin or vamifeport, labeled the residues interacting in hepcidin in Figure 2—figure supplement 5 and introduced the novel panel Figure 4—figure supplement 2D to illustrate common interactions in both molecules. As described in the results, the interactions in the N-pocket strongly overlap whereas there is less overlap with interactions in the C-pocket. The fact that both molecules bind distinct protein conformations complicates a detailed comparison particularly with respect to the energetics of binding.

10) Figure 4 supplement 3B should include the interactions, possibly as a schematic.

We have added both corresponding schemes as panels to Figure 4—figure supplement 3D and F.

11) Lines 268-271 refer to the analysis of the metal binding site and speculation that the conformation stabilized by the Sy3 binder precludes the simultaneous binding of both vamifeport and metal ion. However, physiologically, can the authors demonstrate whether vamifeport prevents Fe binding?

We currently do not have definitive experimental evidence on the question concerning a competition of vamifeport and metal ions. The (partial) inhibition of Fe^2+^ transport in the presence of vamifeport was demonstrated previously (Manolova et al., 2019). However, whether this inhibition is predominantly an effect of stalling the transporter in one conformation thus preventing transport by an alternate exchange mechanism or its removal from the plasma membrane is currently unclear. Experiments addressing a potential competition between metal ions and vamifeport carried out in the context of this study were inconclusive and were thus not described in the manuscript. Moreover, even in case vamifeport prevents binding to S2, as suggested by our data, S1 would still be accessible.

12) Lines 315-316 refer to the two major advances reported in this study: the structures of FPN in the occluded state and the subsequent complex and functional characterization of vamifeport binding. However, it is unclear to what extent the structures are related to physiological relevant states, given that the sybodies appear to stabilize the metal ion binding site in a conformation that cannot now bind metal ions. This section needs more explanation.

We have now removed any claim that the occluded conformation would be on the pathway of conformational changes between inward and outward-facing states. We have discussed this explicitly in a paragraph in the discussion, see response to point 1.

13) Lines 329 – 331, we think the first study to reference the role of helix 7 in the MFS as the 'gating helix' was the 2003 GlpT paper (10.1126/science.1087619), followed by the FucP (10.1038/nature09406) and POT (doi.org/10.1038/emboj.2010.309) studies.

We have included these references.

14) Lines 332-334 – these lines describe what appears to be a significant mechanistic feature, that vamifeport binding causes TM7 to rigidify, which might also be a component of the inhibition mechanism. Why is this not clearly described in the Results section?

While we clearly observe an improvement in the density in the loop preceding helix α7b, this loop is already predominantly structured in absence of the inhibitor (likely as a consequence of its interaction with Sy3, see Figure 4—figure supplement 1C, D). We have explicitly mentioned that the binding of vamifeport further stabilizes this loop region, thus likely also the observed occluded conformation and also referred to a plausible contribution to the mechanism of inhibition. However, we also emphasize that the nature of this inhibition and whether it primarily affects a slowdown of transport or an internalization of the transporter has not yet been clarified.

15) Lines 339 – the authors appear to focus a great deal of comparative analysis between FPN and the GLUT sugar transporters. Why? It would seem to me a more similar system would be by the multidrug exporters or peptide transporters, which also contain multiple binding sites. The link between FPN and GLUTs is not well described to make this comparison obvious.

The comparison with GLUTs is motivated by the fact that for this transporter outward-facing and occluded states are available where the protein appears to undergo a generally similar conformational change as observed in FPN. In our revision, we have removed the paragraph comparing the substrate binding features of GLUTs and FPN and the corresponding panels in Figure 6—figure supplement 1, but we have kept the comparison related to the movement of α7b in the transition towards an occluded conformation.